

**Characterization of Arctic mixed-phase cloud properties at**
**small scale and coupling with satellite remote sensing.**

Guillaume Mioche[1,2], Olivier Jourdan[1,2], Julien Delanoë[3], Christophe Gourbeyre[1,2], Guy
Febvre[1,2], Régis Dupuy[1,2], Frédéric Szczap[1,2], Alfons Schwarzenboeck[1,2], and Jean-François
Gayet[1,2].
[1] Université Clermont Auvergne, OPGC, Laboratoire de Météorologie Physique, F-63000 Clermont-Ferrand,
France
[2] CNRS, UMR 6016, LaMP/OPGC, BP80026, 63177 Aubière, France
[3] Laboratoire Atmosphère, Milieux et Observations Spatiales, UVSQ/CNRS/UPMC-IPSL, 78035,
Guyancourt, France
*Correspondence to:* Guillaume Mioche (g.mioche@opgc.univ-bpclermont.fr)
**Abstract.** This study aims to characterize the microphysical and optical properties of ice crystals and
supercooled liquid droplets within low-level Arctic mixed-phase clouds (MPC). We compiled and analyzed
cloud in situ measurements from 4 airborne campaigns (18 flights, 71 vertical profiles in MPC) over the
Greenland Sea and the Svalbard region. Cloud phase discrimination and representative vertical profiles of
number, size, mass and shapes of ice crystals and liquid droplets are assessed. The results show that the liquid
phase dominates the upper part of the MPC with high concentration of small droplets (120 cm$^{-3}$, 15 µm), and
averaged LWC around 0.2 g.m$^{-3}$. The ice phase is found everywhere within the MPC layers, but dominates the
properties in the lower part of the cloud and below where ice crystals precipitate down to the surface. The
analysis of the ice crystal morphology highlights that irregulars and rimed are the main particle habit followed by
stellars and plates. We hypothesize that riming and condensational growth processes (including the Wegener-
Bergeron-Findeisen mechanism) are the main growth mechanisms involved in MPC. The differences observed
in the vertical profiles of MPC properties from one campaign to another highlight that large values of LWC and
high concentration of smaller droplets are possibly linked to polluted situations which lead to very low values of
ice crystal size and IWC. On the contrary, clean situations with low temperatures exhibit larger values of ice
crystal size and IWC. Several parameterizations relevant for remote sensing or modeling are also determined,
such as IWC (and LWC) – extinction relationship, ice and liquid integrated water paths, ice concentration and
liquid water fraction according to temperature. Finally, 4 flights collocated with active remote sensing
observations from CALIPSO and CloudSat satellites are specifically analyzed to evaluate the cloud detection
and cloud thermodynamical phase DARDAR retrievals. This comparison is valuable to assess the sub-pixel
variability of the satellite measurements as well as their shortcomings/performance near the ground.
**1  Introduction**
The Arctic region is more sensitive to climate change than any other region of the Earth (**Solomon et al., 2007**).
Clouds and particularly low-level mixed-phase clouds related processes have a major impact on the Arctic
surface energy budget (**Curry, 1995; Curry et al., 1996; Morrison et al., 2011**). Observations suggest that
boundary layer mixed phase clouds (MPC, mixture of liquid droplets and ice) are ubiquitous in the Arctic and
persist for several days under a variety of meteorological conditions (**Mioche et al., 2015; Morrison et al.,**



**2012; Shupe et al., 2011; Shupe and Intrieri, 2004**). They occur as single or multiple stratiform layers of
supercooled droplets near the cloud top from which ice crystals form and precipitate. These clouds and
especially those including liquid layers have a large impact on the surface radiative fluxes and Arctic climate
feedbacks (**Kay et al., 2012; Kay and Gettelman, 2009**). The strong impact of MPC on the energy budget
stems from their persistence and microphysical properties which result from a complex web of interactions
between numerous local and larger scale processes that greatly complicate their understanding and modeling
(**Klein et al., 2009; Morrison et al., 2012**).
However, major uncertainties surround our knowledge of the interactions and feedbacks between the physical
processes involved in their life cycle. This complexity reflects in the large discrepancies of the cloud related
processes representation in numerical models, which in turn impacts their predictive capability in the Arctic. For
instance, Global Climate Models (GCM) tend to underestimate the amount of liquid water in MPC (**Komurcu et
al., 2014**). Therefore, the representation of ice formation and growth processes and their interactions with the
liquid phase (liquid/ice partitioning, Wegener-Bergeron-Findeisen process for example) has to be improved, as
already shown in previous modelling studies (**Prenni et al. (2007)** or **Klein et al. (2009)** among others). The
quantification of climate effects is also hampered by difficulties translating observational characterization into
realistic representations in models at all scales. Among the various cloud properties which need to be more
accurately described, the cloud thermodynamic phase is a parameter of primary importance. The standard
assumption in climate models is that liquid and ice are uniformly mixed throughout each entire model grid box
(**Tan and Storelvmo, 2016**). However some field measurements (see among others **Korolev and Isaac (2003)**)
suggest that different pockets of solely water or ice in mixed phase regions coexist. This has consequences on
how processes like the Wegener-Bergeron-Findeisen process should be parameterized in large scale models. The
spatial scale of mixing can affect the longevity, the precipitation formation and the radiative properties of the
clouds. However, understanding and measuring the spatial phase distribution in low level arctic mixed-phase
clouds remains a challenge.
The recent development of ground based stations (Barrow, EUREKA, NY-Alesund among others) and
spaceborne remote sensing observations (for instance lidar and radar onboard CALIPSO and CloudSat
respectively) allow today reliable studies of Arctic cloud phase variability at a regional scale (**Dong et al., 2010;
Kay and Gettelman, 2009; Liu et al., 2012; Shupe et al., 2011**).
Remote sensing observations from space performed by active instruments onboard CALIPSO (Winker et al.,
2003) and CloudSat (**Stephens et al., 2002**) satellites as a part of the A-Train constellation provide a unique way
of characterizing Arctic cloud vertical properties. CALIPSO is equipped with the Cloud-Aerosol Lidar with
Orthogonal Polarization lidar at 532 and 1064 nm (CALIOP), an imaging infrared radiometer (IIR) and a wide
field camera (WFC). CloudSat is equipped with a 95 GHz cloud profiling radar (CPR). However, the cloud
phase must be assessed prior to the retrieval of the cloud microphysical properties. Moreover, the definition of
the cloud thermodynamic phase strongly depends on the measurement technique and the observation scale.
Thus, it appears relevant to investigate the horizontal and vertical distribution of ice crystals and liquid water
droplets as well as the scale dependent liquid-ice partitioning for different observational techniques. However,
since the remote sensing retrieval algorithms and products rely on indirect measurement techniques involving
hypothesis, they need to be validated (**Cesana et al., 2016; Mioche et al., 2010**). They also provide cloud
properties typically averaged over one kilometer, which may be insufficient to study cloud processes at a



microphysical scale. **Cesana et al. (2016)** showed for example that cloud detection and phase retrieval product
from CALIOP lidar measurements depend strongly on factors such as horizontal and vertical data averaging.
Additionally, although space remote sensing measurements present the great advantage to cover the almost entire
Arctic region, they suffer of inherent shortcomings at low altitude levels (**Blanchard et al., 2014; Liu et al.,**
**2017; Marchand et al., 2008**). In situ and ground based remote sensing measurements may fill these gaps by
providing a detailed characterization of cloud microphysical properties at low levels. In addition, in situ
observations are based on direct measurement techniques and can provide data at a higher spatial resolution
(generally < 100m). Numerous previous studies dedicated to the assessment of the microphysical properties of
Arctic clouds are based on *in situ* measurements (**Avramov et al., 2011; Gayet et al., 2009; Rangno and**
**Hobbs, 2001; Verlinde et al., 2007**). But these works focused mostly on case studies. A few studies aimed to
merge several *in situ* datasets to provide a statistical analysis and representative description of mixed phase cloud
properties. Additionally, most of these studies concerned the Western Arctic region (**McFarquhar et al., 2007**).
The present study provides statistics of liquid and ice properties of Arctic MPC from *in situ* data collected in
single layer MPC during several airborne campaigns in the region of Spitzbergen/Greenland Sea between 2004
and 2010. Vertical profiles of liquid and ice properties, as well as parameterizations are presented. The main
objective is to better understand the processes involved in Arctic low-level MPC life cycle at small scale and
improve cloud parameterizations for modeling and remote sensing algorithms. The results will complement
previous works concerning Arctic clouds characterizations performed in Western Arctic.
However, as *in situ* measurements remain very local in time and space, their representativity must be first
established. In **Mioche et al. (2015)** we have investigated the spatial and seasonal variability of MPC properties
using CloudSat and CALIPSO spaceborne observations. The study showed a large occurrence of MPC all year
long both over the whole Arctic and the Svalbard regions. It was clearly evidenced that the Svalbard region, due
to its specific location near the Atlantic Ocean, presents a larger occurrence of low level MPC compared to the
averaged Arctic. These results enabled us to demonstrate that the vertical structure of the clouds sampled during
our airborne campaigns are rather well representative of the cloud profiles observed from space in the Svalbard
region. This conclusion supports the significance of coupling in situ measurements with spaceborne observations
to evaluate and validate remote sensing algorithms and retrieval products in the Svalbard region. This objective
may be achieved by means of flights performed in time and space co-localisation with the satellite tracks (**Gayet**
**et al., 2009; Mioche et al., 2010**).
This study combines two objectives: (i) we aim at improving the description of mixed phase clouds at low level
altitudes and (ii) to link large and small scale observations (space remote sensing and in situ measurements
respectively). The description of the field experiments, instrumentation and datasets will be made in section 2.
Section 3 will present and discuss the vertical profiles of microphysical properties of the low-level MPC. Key
parameterizations useful for modeling or remote sensing will be proposed in section 4. Finally, the coupling of in
situ and remote sensing observations will be achieved in section 5 where the validation of remote sensing
retrieval products in terms of cloud detection and cloud thermodynamical phase will be achieved.
**2    Field experiments, airborne measurements and meteorological situations**





### 2.1 Airborne campaigns

This study is based on *in situ* data collected in single-layer mixed-phase clouds (MPC) during four airborne
international campaigns organized in the Arctic region, namely:
(i) the Arctic Study of Tropospheric Aerosols, clouds and Radiation experiments (ASTAR, **Jourdan et al.,**
**2010**; **Gayet et al., 2009**) which took place in Spitzbergen (Longyearbyen, Norway, 78° N, 15° E) in April 2004
and April 2007 (hereafter labeled AS04 and AS07). The Polar-2 aircraft operated by AWI (Alfred Wegener
Institute) was flown during these two experiments;
(ii) the Polar Study using Aircraft, Remote Sensing, Surface Measurements and Models, of Climate, Chemistry,
Aerosols, and Transport (POLARCAT, **Delanoë et al., 2013**), which was carried out in northern Sweden
(Kiruna, 68° N, 20° E) in April 2008 (hereafter PO08) during the International Polar Year. Measurements were
performed onboard the French ATR-42 aircraft of SAFIRE (Service des Avions Français Instrumentés pour la
Recherche en Environnement);
and (iii) the Solar Radiation and Phase Discrimination of Arctic Clouds experiment (SORPIC, **Bierwirth et al.,**
**2013**), in the Spitzbergen region in May 2010 (hereafter SO10) with the AWI Polar-5 aircraft.
All the scientific flights, in cloudy environment, related to these four campaigns were carried out above open sea
in the Arctic Greenland sea region as displayed on Fig. 1. The flights during ASTAR and SORPIC field
experiments covered latitudes ranging from 75° N to 79° N while flights during POLARCAT campaign were
performed between 70° N and 73° N. Moreover, the data were all collected at a similar period of the year: i.e.
during Spring (April and May).
For this study, we have selected the measurements corresponding to ascent and descent flight sequences into
single-layer MPC as the main objective is to study the vertical partitioning of ice and liquid thermodynamical
phases. Our dataset consists of 71 cloud profiles (see Table 1) representing more than 21 000 measurement
points at 1Hz (350 minutes of cloud observations), spread over 18 flights performed above arctic open sea water.
Four flights were successfully collocated with the ground tracks of the CALIPSO and CloudSat satellites.

### 2.2 In situ instrumentation


A similar in situ instrumentation was mounted on the three aircraft: Polar-2, Polar-5 and ATR-42. The same data
processing was used in order to derive the cloud microphysical parameters (at a same scale: i.e. ~ 100 m)
presented in this study. A coherent cloud data set has been obtained in order to provide a representative statistical
description of the properties of Arctic mixed-phase clouds sampled over the Greenland Sea during Spring.
The in situ instruments used in this study for the cloud properties assessment are the following:
- the Cloud Particle Imager (CPI, **Lawson et al., 2001**), captures cloud particle images on a 1024x1024
pixels CCD camera with a pixel resolution of 2.3 μm and with 256 grey levels. At least 5 pixels are
necessary to identify a cloud particle, so the particle sizes derived from the CPI range from 15 μm to
around 2 mm. The images are processed using the software developed at the Laboratoire de
Météorologie Physique (LaMP, **Lefèvre, 2007**) based on the original CPIView software (CPIView,
2005, **Lawson et al., 2001**; **Baker and Lawson, 2006**). In particular, it provides particle size



distribution (PSD) and derived parameters (particle concentration, effective diameter, extinction
coefficient and ice water content) as well as a particle habit classification. The data processing method
used to derive the extinction coefficient ($\sigma$) and the Ice Water Content (IWC) is described in the
Appendix A.
- the PMS Forward Scattering Spectrometer Probe (FSSP-100, **Baumgardner et al., 2002; Knollenberg,**
**1981**) provides the droplet size distribution from 3 to 45 µm. The derived parameters from the PSD are
the droplet concentration, the effective diameter, the extinction coefficient ($\sigma$) and the liquid water
content (LWC).
- the Polar Nephelometer (PN, **Gayet et al., 1997**) measures the scattering phase function of an ensemble
of cloud particles (either droplets, ice crystals or a mix), from a few micrometers to about 800 µm.
These measurements are useful to identify spherical from non-spherical particles and thus discriminate
the dominant cloud thermodynamical phase. The extinction coefficient and the asymmetry parameter
(g) are calculated following the methodology presented in **Gerber et al. (2000)**.
- the Nevzorov probe **(Korolev et al., 1998)** uses the hot-wire technique to retrieve the liquid water
content and the total water content. Note that the Nevzorov data are only used to determine liquid water
content during ASTAR 2004 because the FSSP-100 was not used during this campaign. The retrieval
method used to determine the Liquid Water Content is described in Appendix A.
The combination of these probes allows cloud particles description from a few micrometers (typically 3 µm) to
about 2 mm. Data are recorded at 1 Hz frequency which corresponds to a spatial resolution of about 100 m
(according to the aircraft speed). The uncertainties and measurement ranges associated to the derived cloud
parameters are summarized in Table A1.
Finally, in situ measurements accuracy may be hampered by the shattering of large ice crystals on the probe
inlets, inducing smaller particle artifact **(Heymsfield, 2007)** leading to an overestimation of small particle
concentration. For example, previous studies of **Field et al. (2003) and Heymsfield (2007)** showed that
shattering effect may induce an overestimation of about 20 % on the bulk properties and a factor 2 or 3 on the
number concentration of ice crystals. Moreover, the recent intercomparison study by **Guyot et al. (2015)** based
on in situ measurements in a wind tunnel experiment showed that the use of the same measurement technic may
lead to large discrepancies on the particle number. Even through no standard method was available during the
campaigns to accurately determine and remove the impact of shattering (designed tips, particle interarrival time
measurement…), a short analysis is described in Appendix B to evaluate the quality of the in situ measurements
and highlight if shattering effect is present.
The three research aircraft measured basic meteorological parameters along the flight track (see **Gayet**
**et al., 2009**). We recall that the static air temperature is calculated with accuracy better than ± 0.5 K. As the
liquid water content remained lower than 0.6 g.m$^{-3}$ during most of the MPC flights, no significant effects on the
reliability of the temperature measurements were observed during cloud traverses. The altitude and geographical
position parameters were measured from the airborne GPS systems with an accuracy of 50 m.






### 2.3 Normalized altitude and meteorological situations


This study is based on a statistical analysis which consists in merging all available MPC data in order to derive
pertinent microphysical parameters in terms of vertical profiles. Hence, since cloud top and cloud base heights
exhibit significant differences according to the considered meteorological situations, the in situ measurements
altitudes are normalized following the method by **Jackson et al. (2012)**. The cloud top and cloud base refer to
the liquid phase layer, i.e. the part of the cloud containing liquid droplets identified from g-values greater than
0.8 (PN measurements, see section 2.4. below). Within the (water) cloud layer (Eq. (1)) and below the cloud
base (Eq.(2)) the normalized altitudes $Z_n$ are the following:

$$Z_n = \frac{z - z_b}{z_t - z_b} \hspace{6cm} (1)$$

$$Z_n = \frac{z}{z_b} - 1 \hspace{6cm} (2)$$

where $Z_n$ is the normalized altitude, Z the altitude of the measurements, $Z_t$ and $Z_b$ the cloud top and base
altitudes respectively. Thus, an altitude of 1 corresponds to the top of the cloud liquid layer and 0 to its base.
Negative values characterize regions of ice precipitation below the cloud layer and the altitude of -1 defines the
ground level according to Eq. (2).
In order to obtain representative statistical results the cloud layers were stratified over 10 levels with intervals of
0.2 (normalized altitude) which each contains around 2000 observations (i.e. about 10% of the data set). Finally,
the vertical profiles of MPC microphysical properties presented in the following are ofbtained by averaging the
in situ measurements over each normalized altitude layer. The profiles are computed for the whole data set and
for each campaign separately in order to better interpret and discuss the results.

The mean vertical profiles of temperature encountered during the four field experiments are displayed on Fig. 2.
The results show that similar profiles are observed for AS04 and SO10 with a well pronounced temperature
inversion (~ -10 °C) whereas PO8 does not exhibit clear temperature inversion. Colder conditions characterize
AS07 with a temperature down to -20 °C at cloud top. During AS07, the prevailing weather situation was
dominated by very cold air outbreaks coming from higher latitudes **(Gayet et al., 2009)**. Table 2 summarizes the
statistics of MPC top and base altitudes, as well as the liquid containing layer thickness for the 71 selected
profiles. The mean cloud top is located around 1200 ± 310 m while the mean cloud base altitude (referring as the
altitude below which liquid phase is no longer present) is 756 ± 283 m. This is consistent with observations
performed in western Arctic where cloud top altitudes lie between 885 m and 1320 m, and cloud base altitudes
between 420 m and 745 m **(McFarquhar et al., 2007)**. Our measurements also indicate that the average layer
thickness spans from 100 m to 950 m with an average of 444 m.

### 237         2.4 Determination of the cloud thermodynamical phase from in situ measurements






The asymmetry parameter (g) derived from PN measurements is used to discriminate cloud thermodynamic
phase. Indeed, in a previous study, **Jourdan et al. (2010)** have shown with a principal component analysis that
g, which is determined according to the Gerber method **(Gerber et al., 2000; Gayet et al., 2002)**, is a reliable
proxy to determine the cloud phase of Arctic MPC. Large values of g (> 0.83) are typical of an ensemble of
particles optically dominated by liquid water droplets where ice crystals do not significantly affect the optical
properties. On the contrary, smaller values of g (< 0.80) are typical of a cloud optically dominated by ice
crystals, with negligible contribution of liquid droplets. For g ranging from 0.80 to 0.83, both liquid droplets and
ice crystals contribute (more or less) to the optical properties. The optical influence of the ice is the greater (i.e. g
decreases) as the concentration and/or the mass of ice particles becomes larger. These results are well illustrated
and discussed by **Febvre et al. (2012)** where PN measurements were combined with FSSP and CPI data.
From this, the liquid droplets properties are determined from FSSP or Nevzorov probe measurements associated
with g-values greater than 0.8 (i.e. indicating a "liquid-containing" phase). On the same way, the ice crystals
properties are determined from CPI measurements associated with g-values less than 0.83 (i.e. indicating an
"ice-containing" phase). Moreover, CPI images identified as spherical droplets are excluded for the
determination of ice crystal parameters. Table 3 summarizes the phase analysis.
In the following the phase discrimination is therefore considered from an optical point of view, contrary to the
work made by **Korolev et al. (2003)** who used the ice water fraction (IWC/TWC) to identify cloud phase.
**3    Small scale properties of liquid droplets and ice crystal particles within MPC**
The purpose of this section is to provide a quantitative assessment of the average microphysical and optical
properties of the water droplets and ice crystals within the MPC cloud layers at a small spatial scale of around
100 m. However, since the vertical profiles presented in this study are performed from aircraft measurements
and correspond to several distinctive clouds, it should be emphasized that they cannot be strictly regarded as
vertical and instantaneous profiles (each ascending or descending flight sequence is generally made in 5-10
minutes), compared to remote sensing measurements which can provide snapshots of a same cloud. The results
are presented for the four airborne campaigns separately. We recall that liquid water droplets/ice crystals
partitioning is based on the asymmetry parameter values derived from the PN measurements (see section 2.4
above).
**3.1  Liquid phase properties**
Figure 3 shows the average vertical profiles, expressed with the normalized altitude reference, of the extinction
coefficient, the number concentration, the liquid water content and the effective diameter (Figs. 3a to 3d)
measured by the FSSP-100 or deduced from the Nevzorov probe and with the condition that g-values from PN
are greater than 0.8. On this figure, each color corresponds to the mean profile of a specific airborne campaign
(AS04 in blue, AS07 in red, PO08 in green, SO10 in orange) while the black curves represent the average over
all campaigns. The averaged vertical distribution of the droplet size distribution for all the campaigns is also
presented on Fig. 3e.



The MPC properties are characterized by increasing values of LWC with altitude. LWC values range between
0.1 at the bottom of the liquid layer and 0.16 $g.m^{-3}$ near cloud top. The concentration of cloud droplets remains
nearly constant throughout the MPC layers with mean values around 120 $cm^{-3}$. However smaller values are
observed near cloud top. While AS04, AS07 and PO08 display similar vertical profiles (with the same trend and
magnitude), clouds observed during SO10 are characterized by larger values of droplet concentration and LWC
(300 $cm^{-3}$ and 0.3 $g.m^{-3}$). For all airborne campaigns, the extinction coefficient profile is correlated with the
LWC measurements indicating that water droplets mainly drive optical properties of upper MPC layers. The
extinction coefficient presents maximum values in the upper part of the cloud (average around 30 $km^{-1}$), and
smaller extinction in the lower part of the liquid layer (down to 15 $km^{-1}$). Finally, the vertical profiles of the
effective diameter (Fig. 3a) and PSD (Fig. 3e) are consistent with the above mentioned statement as the diameter
is proportional to the ratio of the LWC to the extinction coefficient. Hence, liquid layers exhibit small droplet
sizes, with a slight increase of the diameter from cloud base to cloud top (from 10 to 15 μm). These liquid water
droplets vertical profiles are in accordance with the observations presented in **McFarquhar et al. (2007)** or
**Lawson et al. (2001)** relative to MPC in the western arctic region.

**3.2 Ice phase properties**


The corresponding ice crystal properties derived from the CPI measurements (and with the condition that g-
values from PN are less than 0.83) are displayed on Fig. 4 using the same representation as the liquid phase. In
the following the ice crystal concentration corresponds to particles larger than 100 μm in order to avoid
shattering artifacts on this parameter (see **Febvre et al., 2012**). The remaining parameters ($\sigma_I$, IWC and $D_{eff,i}$)
take into account all CPI images, except those identified as liquid droplets. Averaged values of ice crystal
concentration ($N_i$) and extinction coefficient ($\sigma_i$) are around 3 $L^{-1}$ and 0.4 $km^{-1}$ respectively. IWC and effective
diameter ($D_{eff,i}$) display mean values from 0.01 to 0.035 $g.m^{-3}$ and from 80 to 130 μm respectively. The mean
profiles of these properties do not present a clear trend since they are not very correlated with the height, except
at cloud top where a decrease down to nearly zero at $Z_n=1$ is observed. This indicates that the cloud top layer is
almost exclusively composed of supercooled liquid droplets and eventually few small ice crystals. These results
corroborate the findings from the ISDAC and MPACE campaigns in Western Arctic (**McFarquhar et al., 2007,**
**2011**).
Near the sea level ($Zn < -0.5$) no general trend can be highlighted since ice crystals properties show a large
variability from one campaign to another.
The particle shape vertical distribution was also investigated based on the CPI images in order to provide an
insight of the main microphysical growth processes occurring in such MPC. Figure 5 displays the particle shape
distributions relative to number and mass concentration with $Z_n$ (Figs. 5a and 5b) and temperature (Figs. 5c and
5d). To this purpose, particle shapes have been automatically classified by the algorithm developed at LaMP (see
details in **Lefèvre, 2007**). However, the resulting classifications were supported by an accurate human-eye
visualization in order to control the results and avoid the main shortcomings linked to the automatic
classification. As indicated above, only particles with size greater than 100 μm were taken into account in order
to avoid misclassification of smaller particles and shattering artifacts.





Our results clearly show that rimed and irregulars ice crystals are the dominant shapes within MPC (up to 80 %
of the total). In particular, irregular particles are encountered in all ranges of altitude and temperature. They
account for 30 % to 50 % of the total number concentration (and between 20 % and 30 % of mass concentration)
depending on the altitude or temperature of the MPC layer. Rimed particles are predominant inside the liquid
containing cloud layer ($0 < Z_n < 1$) with a contribution up to 40 % in number (60 % in mass) where low
temperatures (below -18 °C) are observed.
An interesting feature is the significant occurrence (around 40 %) of ice crystals with a predominant a-axis
growth at all cloud levels. Indeed, plates, sideplanes and stellars are the dominant habits among the regular
shapes regardless of the cloud layer altitude. Below the cloud ($Z_n < 0$), precipitating ice crystals are characterized
by a mass concentration dominated by rimed particles and by large number concentration fraction of irregular ice
crystals.
Over all, these results agree with the ones presented in **McFarquhar et al. (2007)** based on in situ observations
of MPC during the M-PACE experiment. They highlighted that small supercooled water droplets dominated the
upper layer of the cloud while larger ice particles were present in the lower part and below the cloud (including
irregular, aggregate or rimed-branched crystals). However, they observed a significant fraction of needles and
columns particles (up to 50% below the cloud) in contradiction with the present study (less than 10 %).
Additionally, our results are not in agreement with the observations described in **Korolev et al. (1999)** where
irregular shaped ice crystals accounted for up to 98 % of the total number of ice particles. This disagreement
could be explained by two reasons. First, **Korolev et al. (1999)** considered a wide variety of clouds sampled in
the Canadian and US Arctic (stratocumulus and cirrus at temperatures ranging from 0 to -45 °C and up to 7.5 km
of altitude) whereas the present study focuses only on MPC in the Svalbard region. Also, the disagreement may
stem from the different image processing used in these studies. For instance, **Korolev et al. (1999)** took into
account particles larger than 40 µm (while a 100 µm threshold was used in our study) and two ice crystal shapes:
pristine (defined as faceted ice single particles) and irregulars were considered (while 10 particles shapes were
accounted for to draw up our results).

### 3.3  Profiles of single scattering properties

The PN scattering phase function measurements (hereafter PhF) provide another way to describe and
discriminate the cloud phase properties of MPC (as demonstrated in **Jourdan et al. (2010)**). Figure 6 displays
the mean PN scattering phase function (Fig. 6a) according to the MPC altitude levels as well as the vertical
profile of the corresponding g-values (Fig. 6b). At cloud top, the PhF is characterized by a rather high scattering
at forward angles (angles lower than 60°), a much lower scattering at sideward angles (60-130°), and enhanced
scattering around 140°. These features are representative of cloud layers dominated by spherical particles
(mainly supercooled liquid droplets), as also indicated by typical g-values greater than 0.83. As $Z_n$ decreases, the
PhF becomes smoother and more featureless as a side scattering enhancement is observed and the 140° peak
attenuates. This behavior can be attributed to the presence of non-spherical ice crystals which increase towards
cloud base (as shown in Fig. 4c). This is in agreement with the continuous decrease of g-values observed from
cloud top (0.84) to cloud base (0.82). Thereby, the increase of ice water fraction is associated with a change of
the behavior of the PhF shape when going deeper into the cloud layer. Figure 6 also shows that the ice phase





region below the cloud layer (-1 < Zn < 0) is characterized by a more flat and featureless PhF with no significant
influence of the altitude, associated with g-values smaller than 0.8. This feature is in agreement with the ice
crystal shapes observed. Below the cloud, a similar shape distribution is observed regardless of the altitude as
shown on Fig. 5 where mainly rimed particles (25% in number, 50% in mass), plates (15% and 10%), stellars
(15% and 20%) and sideplanes (5% and 10%) are present. It is thus clearly shown that the PhF is related to
specific microphysical properties encountered at different cloud levels. These observations corroborate that the
PhF can be regarded as an accurate signature of the main microphysical properties observed in the MPC layers
particles.


**3.4  Discussion on statistical vertical profiles**

The quantitative estimates of the separate properties of droplets and ice crystals may give an insight on the
microphysical processes occurring in MPC. These processes are involved in the MPC life cycle, in particular to
maintain the coexistence of liquid droplets and ice crystals, leading to its persistence (**Morrison et al., 2012**).
More specifically, the increase of droplet size and LWC observed in the vertical profiles is consistent with a
condensational growth process within the liquid phase. The slight decrease on LWC and number concentration
observed at cloud top may be due to turbulent mixing effect (**Korolev et al., 2015**). The analysis of the vertical
profiles of ice properties and ice crystal shapes (cf. Fig .5) shows that the presence of pristine particles, mainly
plates and stellars could be linked to a ice crystal growth by vapor deposition including Wegener-Bergeron-
Findeinsen process (WBF, **Bergeron, 1935; Findeisen, 1938; Wegener, 1911**) when liquid droplets are present
(into the cloud layer). The riming process is also very effective regarding the large contribution of rimed
particles. The large presence of irregular particles is in agreement with the previous studies from **Korolev et al.**
**(1999)** and **McFarquhar et al. (2007)** and suggests that aggregation growth processes, or a combination of
several growth mechanisms are involved. This also indicates that turbulence or mixing into the cloud may have
an important influence by redistributing the precipitating ice crystals in the upper cloud levels.
Theoretic adiabatic LWC has also been determined and compared to the observed values to evaluate the
influence of turbulence or mixing effects on LWC as well as the efficiency of ice growth by WBF process or
riming processes. The profiles of the adiabatic ratio (the ratio of the adiabatic LWC to the observed LWC) are
displayed on Fig. 7 and exhibit subadiabatic values for all campaigns. This means that processes responsible for
a decrease of LWC compared to the adiabatic prediction are prevalent. In particular, this strengthens the
assumption that a turbulent entrainment of dry air, resulting in the evaporation of liquid droplets, may occur at
cloud top. Moreover, this confirms that the WBF and riming processes are efficient and responsible for the
decrease of LWC compared to adiabatic values.
Finally, the analysis of the vertical profiles of microphysical properties from a statistical point of view in the
present study is coherent with the findings of previous works on single case studies (**Avramov et al., 2011;**
**Gayet et al., 2009; Rangno and Hobbs, 2001** among others)

However, Figs. 3 and 4 also showed that significant differences in cloud vertical profiles could appear from one
campaign to another. Our analysis is twofold:



First, the SO10 profiles display larger liquid droplet concentration, extinction coefficient and LWC values (~300 cm$^{-3}$, 60 km$^{-1}$ and 0.3 g.m$^{-3}$ respectively) compared to AS04, AS07 and PO08. At the same time, the ice crystals IWC and effective diameter (< 0.01 g.m$^{-3}$ and < 50 μm respectively) are very low compared to AS04, AS07 or PO08. Therefore, the ice crystals are too small to efficiently consume liquid droplets by WBF or riming processes (**Pruppacher and Klett, 1978**), explaining the prevalence of the liquid phase. The adiabatic ratio on Fig. 7 confirms this assumption where larger values are encountered for SO10. Indeed, a large adiabatic ratio denotes that processes for the depletion of liquid droplets (mainly riming or WBF) are not efficient, or relatively less efficient than in the other situations.

Another reason explaining the large droplet concentration could be a change in the aerosol loading (larger aerosol concentrations induce larger droplet concentrations). To investigate this assumption, aerosols number concentrations are analyzed. Since there were no airborne *in situ* measurements of aerosols during the AS04, AS07 and SO10 campaigns, aerosol measurements from Zeppelin Mountain ground station (475 m above sea level, DMPS instrument, D > 10 nm) are considered. On Fig. 8a, the averaged aerosol concentrations for each campaign and corresponding to the time of the selected flights are displayed. During PO08 campaign, aerosol *in situ* measurements were performed onboard the ATR-42 (CPC3010 instrument, D > 10 nm). They indicate aerosol concentrations close to 120 cm$^{-3}$ for the four selected situations. Moreover, the 6 days backward trajectories starting at 500 m and 1000 m altitude at the time and location of the flights have been computed (not shown here) from the NOAA HySPLIT model (Hybrid Single-Particle Lagrangian Integrated Trajectory model, **Draxler and Rolph, 2003**). This gives an insight on the origin and path of the air masses sampled, and may help to explain the discrepancies observed on cloud properties. Figure 8a shows a clean atmosphere with low aerosol concentrations (less than 300 cm$^{-3}$ in average) However, SO10 values present a larger variability compared to AS04, AS07 or PO08. Indeed, among the 5 situations selected during SO10, two of them present high aerosol concentrations (473 cm$^{-3}$ and 393 cm$^{-3}$) and the 3 remaining present lower values (184 cm$^{-3}$, 94 cm$^{-3}$ and 190 cm$^{-3}$).

For all the 18 situations, the backward trajectories indicate that the air masses came mainly from the North or clean areas (sea ice or open water from Arctic Ocean and Greenland Sea). However, some air masses travel over more polluted regions. This is the case for the two situations of SO10 where high aerosol concentrations are observed as the air mass passed over the polluted Taimyr region in Northern Russia (Fig. 8b). So, for these cases, the large aerosol concentrations are in agreement with the backward trajectories and could be responsible for the larger number of droplets observed during SO10. This high number of droplets may reduce the riming process (aerosols indirect effect), explaining the low values of IWC, $\sigma_{ice}$ and $D_{eff,i}$. Consequently, the liquid droplets are not consumed by the ice crystals, and contribute to the large observed values of LWC. This is in agreement with the previous studies of **Lance et al. (2011) and Rangno and Hobbs (2001)** who highlighted the indirect effect of aerosols on MPC. They showed that "polluted" MPC present higher droplet concentrations and less large ice precipitating particles than "clean" MPC.

However, aerosol concentration measurements have to be taken with care since they were not carried out directly at the flight location. Moreover, the physical and chemical properties of aerosols as well as their CCN and IN ability are needed to fully investigate the influence of aerosols on MPC properties.



Finally, the AS07 vertical profiles clearly showed higher values of ice crystals properties compared to the other
campaigns. Backward trajectories are characterized by air mass origins in clean regions. This is supported by
Mount Zeppelin measurements showing low aerosol concentrations of approximately 200 cm$^{-3}$ (Fig. 8a).
Moreover, the temperatures recorded for this campaign are very cold, with for example a cloud top temperature
frequently below -20 °C. Thus, this environment is more favorable for the growth of ice crystals than AS04,
PO08 or SO10. Only one situation presents large aerosol concentration (400 cm$^{-3}$), but it seems to have no
influence on cloud droplet properties (whereas it was the case for SO10) since no difference was observed
between this situation and the rest of AS07 in terms of vertical profiles of ice and liquid properties. So, it
suggests that the influence of the temperature prevails.

Measurements of key parameters are obviously missing in the present study to accurately quantify the
mechanisms responsible for the formation and growth of droplets and ice crystals within MPC. In particular, the
measurements of ice nuclei (IN) properties are needed to make an accurate ice closure (and quantify for example
the secondary ice production process). A better characterization of cloud dynamics, with accurate high spatial
resolution measurements of vertical velocities into and around the MPC would also be necessary. Accurate
humidity measurements would also be needed to better identify condensational growth of ice crystals (WBF
process or direct condensation of water vapor on ice, as described by **Korolev (2007)**) and resolve the issue of
turbulence and mixing at cloud edges and into cloud. All these parameters are of primary importance to constrain
our assumptions on the microphysical processes. At last, coupling the present results (and further observations
with new parameters and improved instrumentation) with modeling is of course the best way to quantify the
impact of each process in the MPC lifetime. But such a work remains beyond the scope of the present study.

**4   Parameterizations of key microphysical parameters**

In section 3, we have shown that *in situ* data provide a detailed characterization of the microphysical and optical
properties of MPC. These measurements can also be used to develop cloud parameterizations and to evaluate
remote sensing retrieval products or modeling outputs. This section focuses on the key properties and hence
parameters which must be better understood and quantified, namely: (i) IWC (and LWC) – extinction coefficient
relationships, (ii) the variability of the ice and liquid water paths, (iii) the temperature dependent ice crystal
concentration and (iv) the liquid water fraction (ratio of LWC over total water content) as a function of the cloud
level or temperature.
The choice of these parameters stems from their importance for modeling and remote sensing, since these
parameters need to be more accurately characterized to improve the output of numerical simulations or to
validate them, and to enhance the reliability of retrieval algorithms (**Morrison and Pinto, 2006**).

**4.1  Ice and liquid water contents and integrated paths**


Linking cloud microphysical and optical properties is an important step in order to model the cloud radiative
properties or to constrain/develop remote sensing retrieval methods. In particular, accurate IWC-extinction
relationships and integrated properties such as ice and liquid water paths are needed to improve the remote





sensing retrieval products and cloud modelling (**Heymsfield et al., 2005; Waliser et al., 2009**). In this section,
we provide such relationships and parameters based on in situ measurements.

Fig. 9a displays the IWC and the LWC measurements as a function of the ice and droplet extinction coefficient
respectively in logarithmic scale with the temperature superimposed in color. The average values of IWC (and
LWC) over 0.2 log($\sigma$) intervals are displayed by the black squares (with the associated standard deviation) in
order to determine the fitting curves (represented by the red lines). Ice crystals and liquid droplets extinction
coefficients are well correlated with their water content counterparts. The correlation coefficients are high (0.88
for ice and 0.90 for liquid) and the IWC-$\sigma$ and LWC-$\sigma$ relationships are almost linear since the exponent of each
fitting equation is close to the unity.
It should also be noted that adding the temperature parameter in the linear fitting did not improve the accuracy of
the parameterizations, contrary to the previous studies of **Heymsfield et al. (2005)**, **Hogan et al. (2006),** or
**Protat et al. (2007, 2016)**. However these previous studies concerned tropical and mid-latitude clouds and
cover a much broader range of temperatures (from 0 °C down to -65 °C, compared to only -24 °C in our study).

Integrated properties such as LWP and IWP are common modeling outputs which have large uncertainties and
variability according to model specifications **(Waliser et al., 2009)**. Moreover a very few previous studies were
devoted to retrieve these properties in Arctic MPC. Since the flight legs selected in our study are limited to
ascending and descending sequences into single-layer MPC, in situ measurements can used to determine IWP
and LWP according to the following equation:

$\text{IWP (or LWP)} = \int_{ground}^{cloud\ top} IWC\ (or\ LWC)(z)dz$           (3)

We recall that ascending and descending flight sequences are obviously not fully vertical and need about 5-10
minutes to be performed (compared to the snapshots performed by remote sensing measurements). Thus, these
integrated properties are considered as quasi-instantaneous.

Figure 9b displays the ice (green) and liquid (blue) water paths as a function of the cloud top temperature (1 °C
intervals). For cloud top temperatures below -20 °C, IWP and LWP reach values around 30 g.m$^{-2}$ and 50 g.m$^{-2}$
respectively. The IWP decreases when the cloud top temperature increases, to very small values at temperatures
above -10 °C. LWP has a different behavior with a maximum reaching 100 g.m$^{-2}$ at -13 °C. These values are
consistent with the main previous studies devoted to Arctic MPC from **(Hobbs et al., 2001; Pinto, 1998; Pinto**
**and Curry, 2001; Shupe et al., 2006**). They reported mean LWP values in the range of 20-70 g.m$^{-2}$, with some
maxima up to around 130 g.m$^{-2}$, and IWP mean values less than 40 g.m$^{-2}$. However, on can note that all these
previous studies concerned MPC in the western Arctic regions (Barrow, Alaska, Beaufort Sea).

**4.2  Ice crystal concentration**


One of the main challenges concerning the life cycle of MPC is the understanding and modeling of the initiation
and the maintenance of the ice phase. In particular, the assessment of IN concentration is of primary importance





and needs to be improve (**Ovchinnikov et al., 2014**). The life cycle of IN particles, in particular their recycling,
may also play an important role in the MPC lifetime (**Solomon et al., 2015**).
Given the temperatures observed in MPC, heterogeneous ice nucleation mechanisms are preferentially involved.
The concentration of large ice crystals (> 100 μm) in particular may be due to heterogeneous ice formation
mechanisms (**Eidhammer et al., 2010; Prenni et al., 2009**). However which process, among deposition,
condensation, immersion or contact freezing, is mainly responsible for the initiation of ice crystals is still under
debate as modeling studies fail to reproduce the observed ice number concentration (**Avramov and Harrington,**
**2010; Fridlind et al., 2007**) among others). This leads to large discrepancies in the modeling of MPC properties
such as ice/liquid partitioning and their radiative impact. Figure 10 shows the maximum number concentration of
ice crystals with size greater than 100 μm as a function of cloud top temperature for each MPC vertical profile
(colored circles). This figure highlights that ice concentration varies almost exponentially (figure is in
logarithmic scale) with the cloud top temperature, with however a large variability**.** Thus, a relationship may be
fitted in order to parameterize ice concentration as a function of temperature in MPC (equation included in Fig.
10), even though the correlation coefficient is not very high (0.43). The parameterization of **Meyers et al.**
**(1992)**, established for contact freezing mode, is also displayed on the Fig. 10, for comparison purposes with the
present study. Our results are in agreement with **Meyers et al. (1992)** parameterization. However, to go further
on this topic of ice nucleation, CCN/IN and humidity measurements are necessary, as well as modeling studies,
which is beyond the scope of this paper.

**4.3  Liquid water fraction**

Finally, since properties of ice and liquid have been separately determined in section 3, the liquid fraction into
MPC can be accurately determined too. The liquid water fraction (hereafter LWF) is defined as the ratio of liquid
water content LWC over the total water content TWC (IWC+LWC).

Figure 11a displays the liquid fraction according to the normalized altitude. For purpose of comparisons, the
parameterization from **McFarquhar et al. (2007)** (hereafter MF07) determined from in situ measurements
during the Mixed-Phase Arctic Cloud Experiment (M-PACE) is displayed on Fig. 11a by the black dotted lines.
Our relationship deviates from that of MF07. They used in situ measurements from 53 profiles in single layer
MPC sampled over Alaska with temperatures ranging from -3 °C to -17 °C. They observed similar number
concentration with smaller ice crystals with mean effective diameters around 50 μm compared to 100 μm in our
study.
Figure 11b shows the liquid fraction according to cloud top temperature. Each point represents the mean value of
the liquid fraction determined for each profile. The error bars corresponding to the standard deviation display
large values around 80 %, and indicate that liquid fraction variability is important. Nevertheless, Fig. 11b shows
that LWF is well correlated with the cloud top temperature. The decrease in LWF associated with a decrease of
temperature is consistent with Fig. 10 which shows that ice number concentration increases for colder
temperatures.
The liquid fraction is also determined at each cloud level as a function of temperature on Fig. 11c (with 1 °C
temperature interval). The same trend as in Fig. 11a is observed. The liquid water fraction increases with



decreasing temperature. The relationship between LWF and T is nearly linear with similar slopes for the 4
campaigns. However, large shifts (discrepancies) are observed from one campaign to another, especially for
AS07 compared to AS04, PO08 and SO10. This shift is clearly linked to the temperature profiles (see Fig. 2).
However, one can note that the results for the PO08 campaign are consistent with the parameterization from
MF07.
In order to compare our results to those of (**Shupe et al., 2006**), we also determined the liquid water fraction in
terms of water paths (LWP/TWP). Fig. 11d shows a rather good agreement between the two path ratios, showing
that IWP dominates in the coldest clouds ($T_{top}$ around -20 °C in average). On the opposite, LWP fraction is more
important in the warmer MPC ($T_{top}$ above -15 °C). However such liquid fraction determination must be taken
with care since it integrates the ice region below the clouds (**de Boer et al., 2009**).

To our knowledge very few previous studies have been undertaken to assess the liquid water fraction in MPC.
Most of them concerned MPC in western Arctic regions only (**de Boer et al., 2009; McFarquhar et al., 2007;**
**Shupe et al., 2006**). Our results show a rather good agreement with these previous works. The proposed
parameterizations are of great importance to accurately constrain the ice/liquid partitioning in the modeling of
MPC. Indeed, parameterizations used in numerical simulations can be very different from one model to another.
For example, the intercomparison work by (**Klein et al., 2009**) involves 26 numerical models. Among them,
some use a T-dependent partitioning scheme for the discrimination of liquid and water phases. But these
schemes lead to very scattered results: at a cloud top temperature of -15 °C for example, the amount of liquid
water varies from 12 % to 83 % according to the scheme used.

**5    Coupling space remote sensing and airborne measurements**

As shown in the two previous sections, in situ measurements can provide MPC properties at high resolution at
low altitudes and close to the ground level. Thus, these measurements are also suitable to evaluate satellite
remote sensing retrieval products. Indeed, it is well known that spaceborne measurements are subject to large
uncertainties at low altitude levels (**Blanchard et al., 2014; Marchand et al., 2008**). The representativity of the
MPC over the Svalbard region has been evaluated in our previous work (**Mioche et al., 2015**) by assessing the
frequency of occurrence of Arctic MPC from space remote sensing. The results provided a better knowledge of
the regional frame in which airborne or ground-based arctic MPC observations were performed. In particular, we
showed that low level MPC are frequent all along the year around the Svalbard region and over the Greenland
Sea. Nevertheless, we also pointed out that the large uncertainties of space remote sensing observations at low
levels near the surface (<2 km) may significantly hamper the cloud occurrence determination (up to 25 %
uncertainty). Accurate profiles of cloud properties at the very low altitude levels are thus needed to complement
and validate the remote sensing observations.

The objective of this section is to link large scale space remote sensing observations of MPC with collocated
small scale in situ measurements. We explore the potential of the aircraft measurements to evaluate satellite
retrieval products in terms of cloud detection and MPC thermodynamic phase. The retrieval algorithm
evaluated in the following is the DARDAR algorithm described in **Ceccaldi et al. (2013) and Delanoë and**





**Hogan (2008 and 2010)**. DARDAR uses the combination of lidar and radar measurements from CALIPSO-
CloudSat to detect clouds and retrieve their phase and properties.
We recall that four flights during the AS07 and PO08 experiments were successfully collocated with the A-Train
track. The DARDAR algorithm was operated (from CALIPSO/CloudSat satellites data) for the cloud/no cloud
detection and the retrieval of the MPC thermodynamical phase. Figure 12 illustrates the vertical profiles of
DARDAR cloud phase product with the cloud type classification. The flight track is superimposed in black lines.
Table 4 summarizes the conditions encountered during the collocated flights, i.e. date and location, time window
Δt (referring to the satellite overpass time) of the in situ measurements used for satellite comparisons and the
subsequent temperature range.

In order to evaluate the DARDAR cloud retrievals, the DARDAR cloud products along the flight track are
compared to the PN asymmetry parameter values. Two methods are used:
(i) DARDAR products are oversampled (DARDAR pixels are split) to match the PN resolution (around 80
m horizontal)
(ii) PN data are projected (and thus averaged) on the same resolution grid as DARDAR products (pixel size
of 1700 m and 60 m horizontal and vertical respectively, corresponding to approximately 17 in situ data
points)
These two methods are considered in order to assess the impact of sampling resolutions on the cloud detection
and cloud phase retrievals. This may also help to evaluate the sub-pixel representativity of the spaceborne
observations.

**5.1    Cloud/no cloud validation**

Cloud detection is first investigated by comparing the DARDAR cloud detection algorithm (i.e. all classes
including a cloud type) along the flight tracks to the in situ PN measurements considered as the cloud/no cloud
occurrence reference. The comparisons are summarized in Table 5 where the statistics of co-occurrences are
determined at the aircraft and satellite spatial resolutions (i.e. with oversampling DARDAR products or
averaging in situ measurements).
The results show a very good agreement between DARDAR and in situ measurements both for cloud and clear
sky cloud detection. At the in situ resolution 91 % of the clear sky events and 86 % of the cloudy pixels are in
accordance with the PN measurements. The effect of the spatial resolution seems to be negligible as 81 %
DARDAR clear sky pixels and 89% cloudy pixels are validated by in situ measurements when comparing at the
satellite resolution. This indicates that the satellite detection of cloudy pixels is consistent with higher spatial
resolution in situ cloud measurements.
The remaining false detections could be explained by the evolution of the cloud structure (cloud top height,
cloud dissipation) during the time delay between the satellite overpass and the aircraft measurements (delay up
to 85 minutes, cf. Table 4). The undefined DARDAR class corresponds to clouds in 60 % or 76 % of the cases.
This result suggests that most of the undefined DARDAR pixels (at least 60 %) correspond actually to cloudy
pixels. In particular, this occurs at low-levels, where DARDAR retrievals are strongly impacted by the
attenuation of the lidar laser beam by liquid layers as well as the contamination by radar ground echoes. This




assumption is confirmed by the results shown on Fig. 12 where the main part of the undefined DARDAR pixels
(brown) is located close to the surface.
**5.2  Cloud phase validation**
In this section, the cloud phase retrieved by the DARDAR algorithm is compared to the cloud phase derived
from the PN in situ measurements. Figure 13 displays the vertical profile with a normalized altitude reference of
the PN asymmetry parameter. The corresponding color coded classification retrieved from DARDAR (cloud
mask) along the flight track is superimposed on Fig. 13. The left and right panels represent the results obtained at
the in situ and DARDAR sampling resolutions respectively.
Near the cloud top ($0.5 < Z_n < 1$), in situ measurements indicate high values of the asymmetry parameter ($g >$
$0.80$) characteristics of liquid or mixed phase layers in accordance with DARDAR "mixing of ice and
supercooled water" class (orange) at both spatial resolutions. However, some points corresponding to DARDAR
"ice class" (light blue) are also present. Most of the low g-values representative of a dominating ice phase ($g <$
$0.8$) is located in the lower part of the cloud layer ($Z_n < 0.5$) and below the cloud ($Z_n < 0$). These values are
mostly linked to the DARDAR "ice class", as well as the "undefined" class (brown) at levels close to the surface
($Z_n < -0.5$). Figure 13 also shows that, to a lesser extent, some DARDAR cloud classes are not in accordance
with the in situ cloud phase discrimination. The "supercooled water" (red) and the "high ice concentration"
(pink) classes do not seem correlated the asymmetry parameter values. One can note that the distribution of
DARDAR cloud classes over the g-values from PN gives similar results whatever the resolution used
(DARDAR or in situ).
Figs. 12 and 13 show that DARDAR correctly retrieves the MPC typical vertical structure from a qualitative
point of view, i.e. mainly supercooled water (red) or a mixed ice and supercooled water particles (orange) at
cloud top, and ice below (light blue).
A more quantitative and statistical approach is provided on Fig. 14 where the frequencies of occurrence of g are
displayed for the three main phase classes derived from DARDAR cloud type classification. In this figure, only
DARDAR cloudy pixels are selected and the four "ice" classes are gathered in a new "ice" class (blue). The
histograms are determined at the in situ resolution (left panel) and at DARDAR resolution (right panel). We
recall that g-values of 0.80 and 0.83 can be chosen to define the boundaries of the ice, mixed and liquid phases
**(Jourdan et al., 2010)**.
First, one can note that the shape of the frequency distribution is very similar for both spatial resolutions. The
histograms of g-values corresponding to the mixing of ice and supercooled water DARDAR phase (orange) are
centered on g-values of 0.85. The "ice" DARDAR class (blue) distribution exhibits two modes: one (main)
around 0.74, and one around 0.84. Finally, the distribution of the supercooled water class is rather flat and
difficult to interpret given the few number of occurrences. Table 6 summaries the statistics of DARDAR cloud
phase validation, based on these histograms. It shows that 61 % observations corresponding to DARDAR "ice"
class are validated by PN data at the in situ resolution (and 60 % at DARDAR resolution). The remaining
DARDAR "ice" pixels are distributed among the "in situ" mixed (15 % - 20 %) and liquid (24 % - 20 %) phases.





Nearly 90 % of the "mixing of ice and supercooled water" DARDAR class pixels (hereafter called mixing class)
are associated with a liquid phase according to PN (g > 0.83). When considering the statistics at DARDAR
resolution this value drops to 78 %. The other pixels are distributed more or less equally among the "in situ" ice
and mixed phase (6 % and 5 % at the in situ resolution and 14 % and 8 % at the satellite resolution). Finally, 67
% (or 40 % at DARDAR resolution) of the DARDAR pixels identified as "supercooled water only" are validated
by the in situ measurements while 24 % (40 %) correspond to an "in situ" mixed phase. The remaining 9 % (20
%) corresponds to an ice phase. However, the "supercooled water only" statistics are not very representative
since the number of pixels is limited as it corresponds to less than 2 % of the DARDAR cloud pixels along the
flight track. (only 91 and 6 points at in situ and spatial resolutions respectively). Moreover, "supercooled water
only" is flagged when the radar signal is not detected (**Ceccaldi et al., 2013**).

In general, the misclassifications of DARDAR pixels could be attributed to differences in the temporal and
spatial resolution between in situ and satellite measurements. Indeed, the in situ data provide an accurate
description of the cloud thermodynamic phase at high spatial distribution which can take into account the small
scale heterogeneities of the liquid and ice occurrences. On the contrary, due to its lower spatial resolution (one
order of magnitude lower than aircraft measurements), satellite products provide more likely an averaged cloud
phase assessment. Indeed, DARDAR retrieval algorithm uses the horizontal resolution of CloudSat (1.7 km) and
the vertical resolution of CALIOP (60 m). Therefore, it is likely that a DARDAR pixel is dominated by ice and
thus identified as ice, but is in reality composed of several small sequences or pockets of ice and supercooled
liquid droplets. This may explain most of the misclassifications of the ice DARDAR pixels in MPC. On the other
side, in mixed cloud layers optically dominated by supercooled water droplets, the PN signal is representative of
the liquid phase whereas the LIDAR/RADAR synergy can detect the presence of a few ice crystals. This could
also explain the mismatch of the in situ and satellite mixed phase class.
The time synchronization issue may also be responsible for the misclassification of 24 % of the DARDAR pixels
of the "ice" class that should belong to the "liquid phase" according to the PN measurements (g > 0.83). Indeed,
the cloud top and base altitudes as well as the cloud layer thickness may vary during the satellite overpass time
and the aircraft sampling time. Physical assumptions used in DARDAR algorithm can also contribute to the
discrepancies. For instance, the algorithm assumes that the supercooled liquid layer thickness has to be less or
equal to 300 m (**Delanoë and Hogan, 2010**). However, the averaged liquid layer thickness from all MPC
sampled for the present study is 444 +/- 211 m (see Table 2 in section 2.3.). This could contribute to the
disagreement between DARDAR ice phase and PN classification as most of the misclassified pixels are located
in the lower part of liquid layer (Fig. 13). Additionally, it is likely that CALIOP laser beam is fully attenuated by
a 300 m liquid layer thickness which translates into the inability of DARDAR to detect the liquid phase beyond
this thickness.

In order to explain the 40 % of misclassifications of DARDAR ice class, we also evaluated separately the four
DARDAR ice sub-classes ("ice only", "spherical ice", and "high IWC", see Table 7). The "ice only" class
represents the main part of the ice classes (around 80 % of the total number of DARDAR ice pixels). 63 % of the
pixels in this class are validated by the PN data at in situ resolution (60 % at the satellite resolution). The
remaining 37 % (40 %) correspond to a mixed phase (16 % - 21 %) or a liquid phase (21 % - 20 %). 47 % to 63



% of the spherical ice class pixels correspond to an in situ ice phase. The remaining 53 % or 37 % misclassified
pixels seem to belong to the liquid phase (43 % - 30 %), and only 10 % (7 %) to the mixed phase. It may stem
from the strong attenuation of the lidar laser beam by the cloud liquid top, leading to a partial detection of the
liquid layer by DARDAR. This feature is well highlighted on Fig. 12, where the spherical ice DARDAR pixels
(dark blue) are almost always located in the vicinity of the "mixing of ice and supercooled water" pixels (orange)
in the cloud layer.
The remaining ice sub-class namely the "high ice concentration" is not in accordance with the dynamical
processes responsible for the formation or maintenance of boundary level mixed phase clouds. This class is more
likely to be involved in strongly convective clouds. These misclassifications indicate that the DARDAR
detection scheme could be improved in presence of low-level clouds. However, we should keep in mind that the
number of these DARDAR pixels is very low and hence not necessarily representative as these two classes
represent only 3 % of the total DARDAR ice pixels along flight tracks).

At last, it is quite remarkable that the results are very similar for the in situ and the satellite resolutions, both in
terms of cloud/no cloud detection and for the cloud phase product. The results highlight a very good
representativeness of the spatial observations concerning the cloud detection. The cloud phase is also well
retrieved, but some issues have been identified regarding the retrieval of the supercooled water phase, which is
mostly confounded with the mixing of ice and supercooled water class. The retrieval of the ice phase also
presents some misclassifications as some part of the retrieved ice pixels is actually liquid water. These issues
may be mainly attributed to the difference in resolution between spatial and in situ observations with a better
ability to detect the heterogeneities (sequences of liquid and ice) of mixed phase at small scale (in situ) compared
to large scale (spatial resolution). These results highlight the importance of a characterization at high resolution
of the cloud thermodynamical phase.

Finally, we recall that the uncertainty on the asymmetry parameter is 4% **(Gayet et al., 2002a)**, and the g-values
thresholds used for the cloud phase discrimination remain empirical. In order to provide a confidence interval of
the validation scores presented in Tables 5, 6 and 7, the validation statistics have been recomputed using g-
values thresholds with +/- 0.01 variability (between 0.79 and 0.81 for the discrimination between ice and mixed
phases, and between 0.82 and 0.84 for the discrimination between mixed and liquid phases). All the validation
scores resulting from these new g-values thresholds have been compared to those in Table 6 and the subsequent
mean variation is estimated to be +/- 10 %.


**6    Conclusions and outlook**

In this study, a characterization of Arctic boundary-layer mixed phase clouds microphysical properties has been
performed. In situ data from 4 airborne campaigns over the Greenland Sea and the Svalbard region are compiled
and analyzed. The data set represents in total 18 flights and 71 vertical profiles in MPC. Cloud phase
discrimination is achieved and vertical profiles of number, size, mass and shapes of ice crystals and liquid
droplets within MPC are determined. Furthermore, 4 flights were collocated with CALIPSO and CloudSat



satellites tracks. The corresponding spaceborne and in situ collocated data are used to evaluate satellite cloud phase retrieval product (DARDAR cloud type) and to fill the gap of spaceborne remote sensing measurement near the surface.

The main conclusions of the present work are summarized as follow:

i) More than 350 minutes of cloud in situ observations have been merged to characterize the arctic MPC microphysical and optical properties. Vertical profiles of liquid droplets and ice crystals properties have been determined separately to allow for an accurate description at high spatial resolution of MPC near the ground. Liquid phase is mainly present in the upper part of the MPC with high concentration of small droplets (120 cm$^{-3}$, 15 µm), and averaged LWC around 0.2 g.m$^{-3}$. Ice crystals are present everywhere in the MPC, but mainly in the lower part, and precipitate down to the surface. The morphology study of ice crystals images showed that irregular and rimed particles prevail over stellars and plates habits.

ii) The vertical profiles of the microphysical properties and the shape distribution can also be used to give an insight of the microphysical processes occurring in MPC. It is likely that adiabatic lifting (condensation) is the main process for liquid droplets initiation and growth, and that evaporation at cloud top due to entrainment of dry air seems to occur. In the cloud layer, where liquid droplets and ice crystals coexist, Wegener-Bergeron-Findeinsen and riming processes are the main mechanisms involved in the ice crystal growth. The large occurrence of irregular particles highlights the role of aggregation and turbulence in the MPC life cycle.

iii) The analysis of the scattering phase function showed a very high correlation between optical properties and liquid to ice fraction within the MPC layers.

iv) The differences observed in the vertical profiles of MPC properties from one campaign to another highlighted that the large number of liquid droplets observed on two situations during SO10 is in part linked to the source and transport of aerosols properties. For this campaign, the large values of droplet number and LWC are associated with very low values of ice crystal size and IWC. On the opposite, the very cold and clean situations of AS07 exhibit large values of ice properties. These results underline the importance of studying aerosols measurements (sources, transport, physical and chemical properties) in connection with the MPC properties to study the cloud-aerosol interactions and improve the understanding of ice and liquid formation processes.

v) Several parameterizations which may be relevant for remote sensing or modeling are proposed. It concerns the determination of IWC (and LWC) – extinction relationships, ice and liquid integrated water paths, the ice concentration and liquid water fraction. Comparisons with the few previous works available in the literature showed a good agreement. Obviously, the





application range of the established relationships is only for arctic MPC and temperature range
between 0 and -23 °C.
vi)    The analysis of collocated in situ and spaceborne observations was considered in order to link
large scale to small scale observations. CALIOP/CloudSat observations processed with the
DARDAR algorithm lead to a good retrieval of the MPC structure i.e. supercooled liquid at
cloud top and ice below. Globally, more than 80 % of the clear sky pixels and the cloudy
pixels retrieved by DARDAR are validated by in situ observations. The analysis pointed out
that a large part of cloudy pixels (around 70 %) near the surface level cannot be detected. This
corroborates the well-known difficulties encountered by space remote sensing near the surface
(lidar laser beam attenuation or radar ground echoes) and already highlighted by previous
studies. These results highlight the need for satellite observations to be completed by
observations near the surface and at a more detailed scale. In situ measurements are thus
excellent candidate to fill the gap of satellite observations.
vii)    The evaluation of DARDAR cloud phase product revealed that the low spatial resolution of
satellite product (1.7 km horizontal) leads to large misclassifications. For example, only 61 %
of ice DARDAR pixels are validated, and the most part of mixing of ice and supercooled water
DARDAR class is actually only supercooled water. Time and space synchronization may also
play a role in these misclassifications, but in a lesser extent. This evaluation work highlighted
the need of accurate MPC properties and profiles near ground. Moreover, this work allowed
the evaluation of the sub-pixel variability of the spatial observations.
This study provided for the first time a statistical analysis of arctic MPC in situ data from 4 airborne campaigns
located in the Svalbard/Greenland sea region.
An accurate characterization of liquid droplets and ice crystals properties separately has been made. However,
accurate measurement of humidity and aerosol (CCN and IN) remains an important lack in order to go deeper in
the analysis of microphysical processes to realize ice and liquid closure and better understand life cycle and
persistence of such particular clouds. Modeling will also help in this task. Finally, by comparison with collocated
spaceborne observations, this study allowed to establish a link between large and small scale observations. This
methodology could be applied to airborne remote sensing observations such the RALI system and for the future
space observations devoted to cloud studies, such the EarthCare mission, planned for 2019.

**Appendix A: Data processing of in situ measurements**

The methodology developed by **Lawson and Baker (2006)** to derive the Ice Water Content (IWC) from 2D
particle images recorded by the CPI instruments is applied (Eq. (A1) below).
$IWC = \dfrac{0.135 \sum_i x_i^{0.793}}{V}$           (A1)



where $V$ is the sample volume and $X_i$ is the mass parameter for each crystal image defined by Lawson and Baker
(2006) as follow:
$$X_i = \frac{A_i \times W_i \times 2 \times (L_i + W_i)}{P_i}$$     (A2)
$A_i$, $W_i$, $L_i$ and $P_i$ are the area, width, length and perimeter of the crystal image $i$.

The extinction coefficient ($\sigma$) and the effective diameter ($D_{eff}$) are determined from CPI and FSSP measurements
as follow:
$$\sigma_{ice \ (or \ liquid)} = 2 \times \frac{\sum_i A_i}{V}$$     (A3)
$$D_{eff,ice \ (or \ liquid)} = C \times \frac{IWC \ (or \ LWC)}{\sigma_{ice} \ (or \ \sigma_{liquid})}$$     (A4)
with constant C = 3000 mm³.g⁻¹ according to **Gayet et al. (2002b)**.
The LWC derived from the Nevzorov probe measurements is calculated according to **Korolev et al. (1998)** :

$$LWC_{Nevzorov} = \frac{P_{LWC} - \left(\frac{P_{TWC} \times \varepsilon_{LWC,i} \times S_{LWC}}{\varepsilon_{TWC,i} \times S_{TWC}}\right)}{L_v \times S_{LWC} \times U \times \left(\varepsilon_{LWC,l} - \frac{\varepsilon_{LWC,i} \times \varepsilon_{TWC,l}}{\varepsilon_{TWC,i}}\right)}$$     (A5)

where $P_{LWC}$ and $P_{TWC}$ are the power supplied to the LWC and TWC sensors to maintain the constant temperature
of the wire.
$S_{LWC}$ and $S_{TWC}$ are the surface of the sensors, $L_v$ is the latent heat of vaporization and U is the true airspeed.
The epsilon terms refer to the collection efficiencies of liquid droplets (l index) or ice crystals (i index) on the
LWC and TWC sensors. These efficiencies are set as follow:

$\varepsilon_{LWC,l}$ = 0.76 : see **Schwarzenboeck et al. (2009)**;

$\varepsilon_{LWC,i}$ = 0.11 : following **Korolev et al. (1998)**;

$\varepsilon_{TWC,l}$ = 1 : according to **Korolev et al. (1998)** for droplets with size around 25 µm;

$\varepsilon_{TWC,i}$ = 1 : following **Schwarzenboeck et al. (2009)**. It should be noticed that taking $\varepsilon_{TWC,i}$ = 3 (as

assumed in **Korolev et al., 2013**) instead of 1 induces an increase of LWC by 10 % only.

The uncertainties associated to the microphysical and optical properties derived from FSSP-100, PN, Nevzorov
and CPI measurements are detailed in **Baumgardner and Spowart (1990), Gayet et al. (2002b), Korolev et al.**
**(1998) and Mioche (2010)** respectively, and are summarized in Table A1.
**Appendix B: Effects of shattering of ice crystals on measurements**



Techniques and methods exist now to avoid or estimate this shattering effect, such as new-designed inlets or
measurements of the particles inter-arrival time **(Field et al., 2003)**, but none of these were available for this
study. However in order to assess the accuracy of the present dataset and highlight a possible impact of
shattering effect, a brief intercomparison of the extinction coefficient from the three data sets was conducted.
Indeed, the extinction coefficient is the only parameter which can be derived by the measurements of the three
probes. Moreover, it is not determined with the same method, since it is calculated from the PSD for the CPI and
the FSSP, and from the scattering phase function for the PN. One more important point is that CPI, FSSP and PN
have all different size inlets (23 mm, 40 mm and 10 mm diameter respectively). So, from these information, we
could assume that, if shattering effect is present on ice particles, its magnitude (i.e. the number of smaller new
artifact particles) would differ from one instrument to another. Thus, the comparison of the extinction coefficient
from CPI, FSSP and PN measurements would highlight such discrepancies.
Figure B1 displays the comparison of the extinction coefficient derived from the PN and from the combination
of the CPI and FSSP for all the in situ data available for this study. Note that the combination of CPI and FSSP
data covers the same size range of the PN. Figure B1 clearly shows that the extinction coefficient measurements
derived from the combination of the CPI and FSSP and the PN are very well correlated (with a coefficient of
0.87) and no significant bias is observed (regression coefficient of 0.98). Thus, since the design of the
instruments and data processing are different for each dataset, these results highlight that the shattering effect is
probably smaller than the measurements uncertainties (25 %, 35 % and 55 % for PN, FSSP and CPI respectively,
see Table A1).
*Acknowledgments*
This research was funded by the Centre National de la Recherche Scientifique – Institut National des Sciences de
l'Univers (CNRS-INSU) and the Expecting EarthCare Learning from A-Train (EECLAT) project. We thank the
Alfred Wegener Institute (AWI) and the Service des Avions Français Instrumentés pour la Recherche en
Environnement (SAFIRE) for the organization of the campaigns and for providing research aircrafts. The authors
acknowledge the NOAA Air Resources Laboratory (ARL) for the provision of the HYSPLIT transport and
dispersion model and READY Web site (http://www.arl.noaa.gov/ ready.html) used in this publication. We
thank Peter Tunved from the Stockholm University for providing via the EBAS database the aerosol data from
Mount Zeppelin station. We thank anonymous reviewers who made important comments which strengthened the
manuscript.

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





**Table 1: Summary of in situ observations concerning Arctic single layer MPC. In brackets: number of flights**
**collocated with CALIPSO/CLOUDSAT tracks.**

| Field experiment | Location [latitude range] | Date | Number of flights in MPC | Number of profiles in MPC | Duration of data (minutes) |
|---|---|---|---|---|---|
| ASTAR 2004 | Spitzbergen (Norway) [76-79]° N | May 2004 | 4 | 7 | 30 |
| ASTAR 2007 | Spitzbergen (Norway) [76-79]° N | April 2007 | 5 (2) | 34 | 173 |
| POLARCAT 2008 | Kiruna (Sweden) [68-73]° N | April 2008 | 4 (2) | 10 | 45 |
| SORPIC 2010 | Spitzbergen (Norway) [75-78]° N | May 2010 | 5 | 20 | 109 |
| **TOTAL** | | | **18 (4)** | **71** | **357** |







**Table 2: Base and top altitudes and layer thickness statistics from the 71 profiles sampled in MPC.**

|  | Mean | Standard dev. | Median | 25th percentile | 75th percentile | Max. | Min. |
|---|---|---|---|---|---|---|---|
| $z_{top}$ (m) | 1200 | 310 | 1200 | 1000 | 1370 | 2120 | 525 |
| $z_{base}$ (m) | 756 | 283 | 700 | 510 | 850 | 1700 | 400 |
| Layer thickness (m) | 444 | 211 | 420 | 270 | 600 | 950 | 100 |







**Table 3: Summary of the method for the assessment of the cloud thermodynamical phase and liquid droplet and ice**
**crystal properties from the combination of PN, CPI, FSSP and Nevzorov probes.**

| | PN g-values | | |
| --- | --- | --- | --- |
| | [corresponding cloud phase] | | |
| | g < 0,80 | 0,80 < g < 0,83 | g > 0,83 |
| | [ice] | [mixed] | [liquid] |
| Instrument [measurement range] | | | |
| FSSP [15 to 45 μm] | NO | YES | YES |
| Nevzorov probe [LWC > 0.003-0.005 g.m-3] | NO | YES | YES |
| CPI [15 μm to 2.3 mm] | YES | YES | NO |






**Table 4: Date, location, time difference between aircraft sampling and satellite overpass (Δt) and temperature range**
**for the 4 flights collocated with CALIPSO/CLOUDSAT satellite tracks.**

| Date | Location | Δt (minutes referring to satellite overpass) | Temperature range (°C) |
|---|---|---|---|
| **7 april 2007** | West of Svalbard, over ocean | - 10 to +15 minutes | -21 to -11°C |
| **9 april 2007** | West of Svalbard, over ocean | -25 to +40 minutes | -22 to -11 °C |
| **1rst april 2008** | North of Sweden, over ocean | -5 to +45 minutes | -20 to -3°C |
| **10 april 2008** | North of Sweden, over ocean | +20 to +85 minutes | -21 to -3°C |






**Table 5: Statistics of DARDAR cloud detection validation.**

| | | PN (reference) | | Total number |
| | | No cloud | Cloud | of points |
| **Resolution** | **DARDAR class** | | | |
| | DARDAR clear sky | 91 % | 9 % | 4840 |
| In situ | DARDAR cloud | 14 % | 86 % | 5245 |
| | DARDAR undefined | 40 % | 60 % | 593 |
| | DARDAR clear sky | 81 % | 19 % | 248 |
| Satellite | DARDAR cloud | 11 % | 89 % | 312 |
| | DARDAR undefined | 24 % | 76 % | 37 |







**Table 6: Statistics of DARDAR cloud phase validation.**

| Resolution | DARDAR retrieval | phase | PN data (reference) | | | Total number of points |
|---|---|---|---|---|---|---|
| | | | Ice phase (0.75 < g < 0.80) | Mixed phase (0.80 < g < 0.83) | Liquid phase (g > 0.83) | |
| In situ | Ice | | 61 % | 15 % | 24 % | 3151 |
| | Mixing of ice and supercooled water | | 6 % | 5 % | 89 % | 1289 |
| | Supercooled water only | | 9 % | 24 % | 67 % | 70 |
| Satellite | Ice | | 60 % | 20 % | 20 % | 186 |
| | Mixing of ice and supercooled water | | 14 % | 8 % | 78 % | 87 |
| | Supercooled water only | | 20 % | 40 % | 40 % | 5 |







**Table 7: Statistics of DARDAR ice sub-classes validation.**

| Resolution | DARDAR ice sub-classes retrieval | PN data (reference) | | | Number of points |
| --- | --- | --- | --- | --- | --- |
| | | Ice phase (0.75 < g < 0.80) | Mixed phase (0.80 < g < 0.83) | Liquid phase (g > 0.83) | |
| In situ | Ice only | 63 % | 16 % | 21 % | 2753 |
| | Spherical ice | 47 % | 10 % | 43 % | 297 |
| | High IWC | 48 % | 18 % | 34 % | 101 |
| Satellite | Ice only | 60 % | 21 % | 20 % | 147 |
| | Spherical ice | 63 % | 7 % | 30 % | 27 |
| | High IWC | 58 % | 17 % | 25 % | 12 |







**Table A1: Uncertainties on cloud properties derived from CPI, FSSP, PN and Nevzorov measurements.**

| Probe<br>[Measurements range] | Number concentration (N) | Extinction coefficient ($\sigma$) | Effective diameter ($D_{eff}$) | Water contents (IWC or LWC) | Asymmetry parameter (g) |
|---|---|---|---|---|---|
| **CPI**<br>[15 µm to 2.3 mm] | 50 % | 55 % | 80 % | 60 % | - |
| **FSSP-100**<br>[3 to 45 µm] | 10 % | 35 % | 4 % | 20 % | - |
| **PN**<br>[< 800 µm] | - | 25 % | - | - | 4 % |
| **Nevzorov**<br>[LWC > 0.003-0.005 g.m$^{-3}$] | - | - | - | 20% | - |







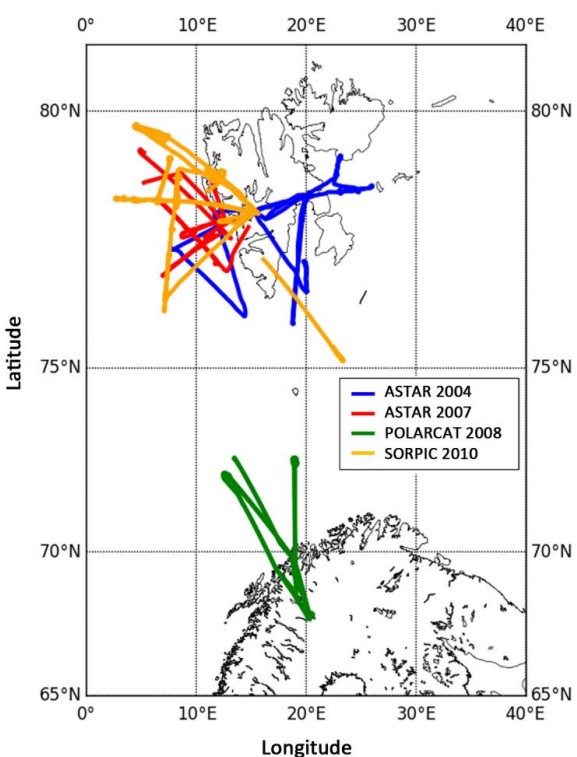

**Figure 1: Flight trajectories related to MPC measurements during the ASTAR, POLARCAT and SORPIC**
**campaigns.**





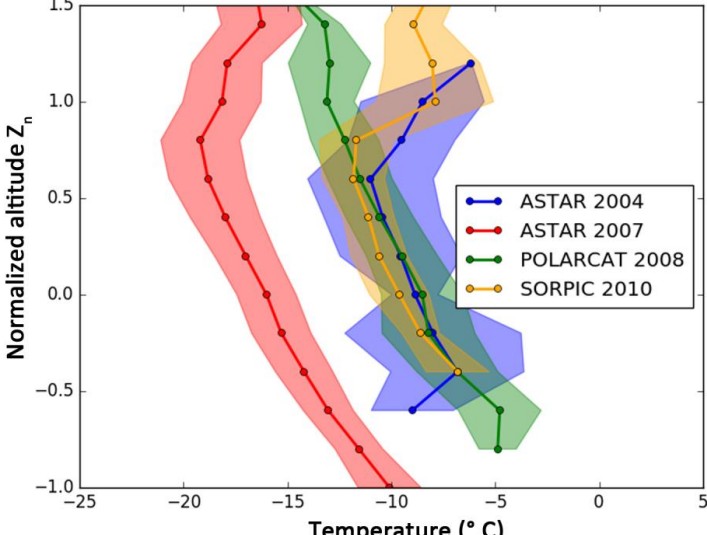



**Figure 2: Vertical profiles (normalized altitude) of the mean temperature for each experiment. Shaded spreads**
**represent the standard deviation.**






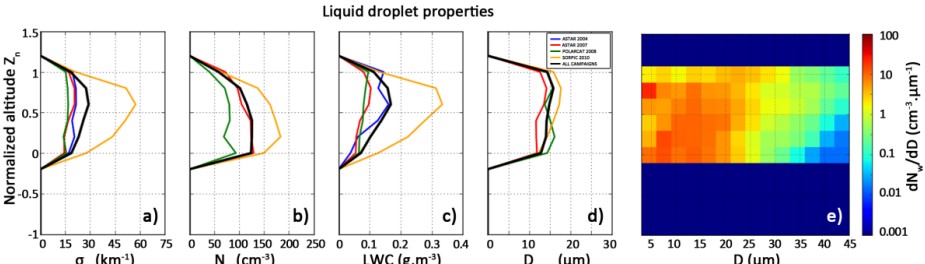

**Figure 3: Vertical profiles (expressed in normalized altitude) of liquid droplets properties from FSSP measurements (3-45 µm size range): a) extinction coefficient, b) droplet concentration, c) LWC, d) effective diameter and e) averaged droplet size distribution for all the campaigns.**





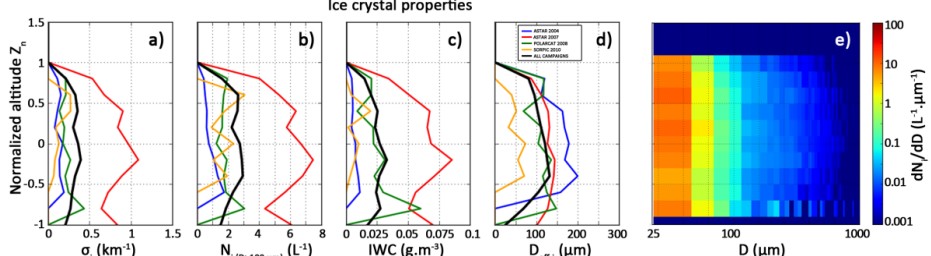


**Figure 4: Vertical profiles (expressed in normalized altitudes) of ice crystal properties from CPI measurements (15 µm - 2.3 mm size range): a) extinction coefficient, b) ice crystal concentration, c) IWC, d) effective diameter and e) averaged particle size distribution for all the campaigns.**




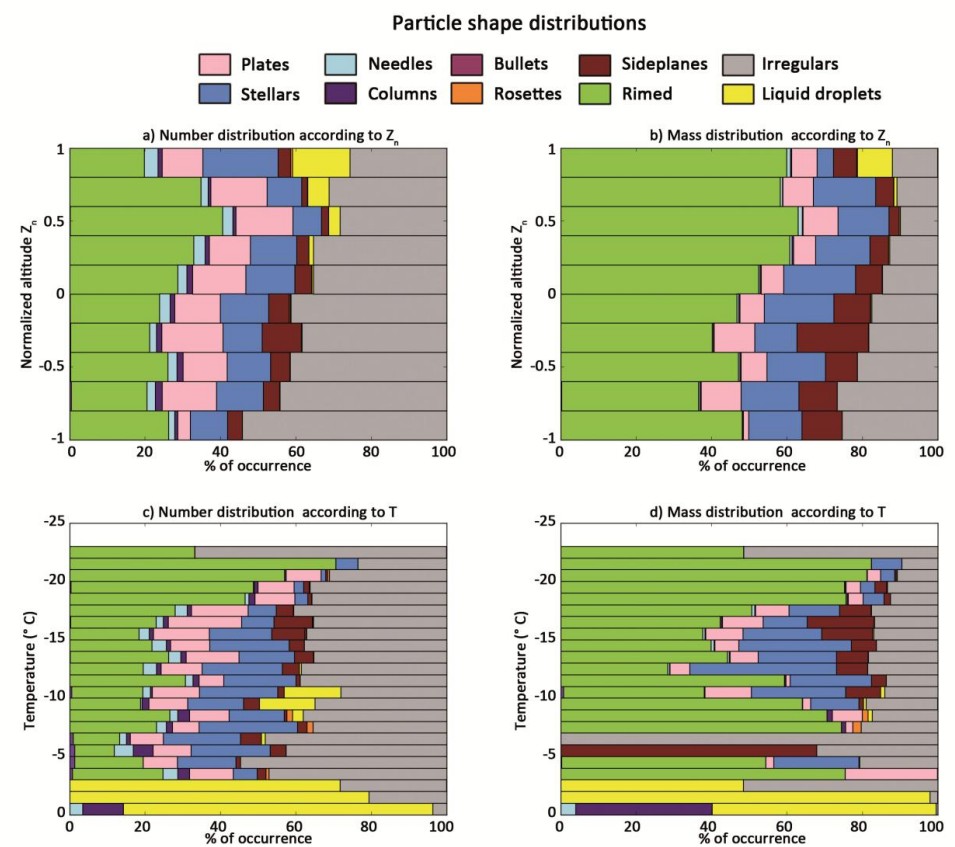

**Figure 5: Vertical profiles of particle shapes (from CPI measurements and for particles larger than 100 μm) according to normalized altitude (top panels) and temperature (bottom panels). Distributions are displayed according to particle number (left panels) and mass (right panels).**





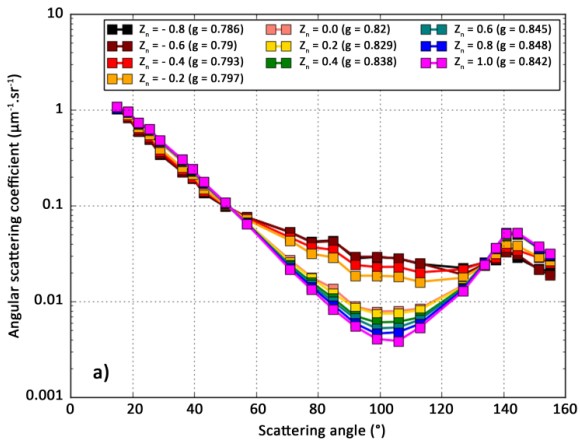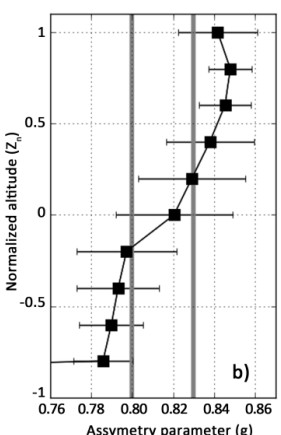



**Figure 6: a): Normalized scattering phase function according to the normalized altitude from Polar Nephelometer**
**measurements (few μm to around 800 μm size range), averaged over all the campaigns. g-values indicate the cloud**
**phase: g < 0.80: ice, 0.80 < g < 0.83: mixed and g > 0.83: liquid. b): Mean vertical profile of asymmetry parameter (for**
**all the campaigns). The grey bars indicate the threshold g-values for the assessment of ice, mixed and liquid cloud**
**phases.**






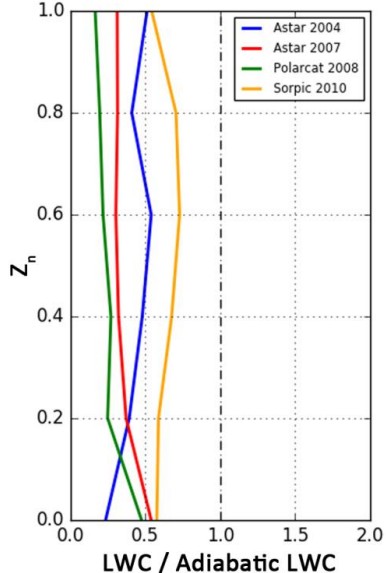

**Figure 7: Vertical profiles of the ratio of measured LWC over theoretical adiabatic LWC.**





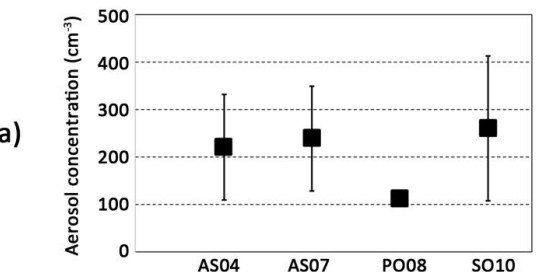

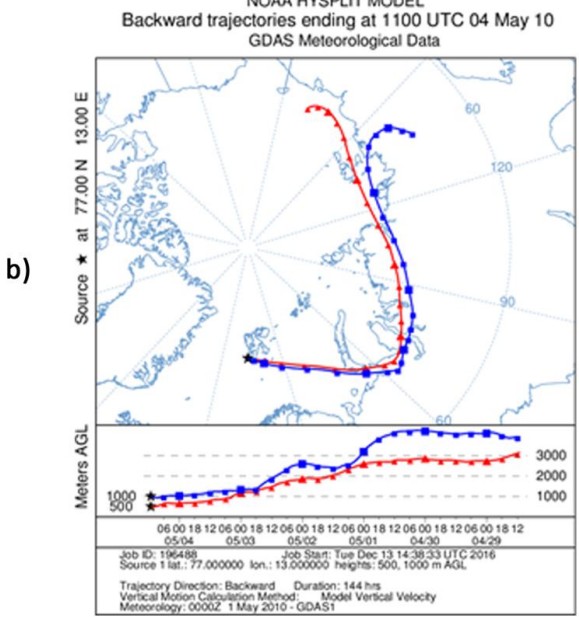



**Figure 8: a) Averaged aerosol number concentrations measured at flight time at the Zeppelin Mountain station for**
**AS04, AS07 and SO10 flights, and onboard the ATR-42 aircraft for PO08. Errors bars displays the standard**
**deviations; b) Backward trajectory from Hysplit model for the 4 may situation (SO10).**






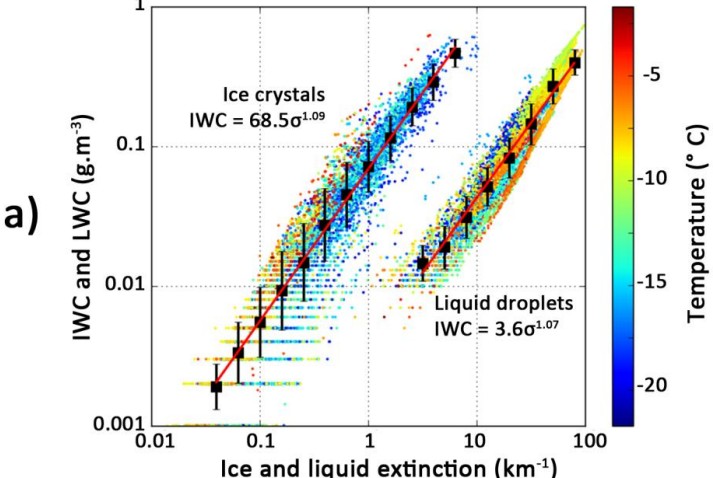

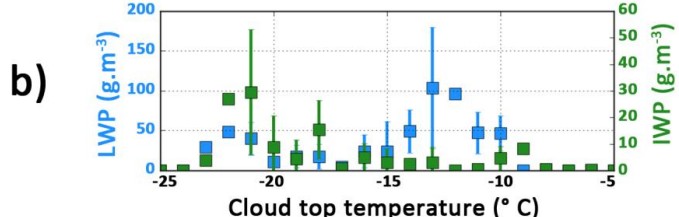

**Figure 9: a) IWC and LWC as a function of extinction coefficient. Color scale is the temperature, black squares**
**represent the values averaged over 0.2 log(σ) intervals and the red lines represent the fittings; and b) ice (green) and**
**liquid (blue) water paths according to the cloud top temperature.**





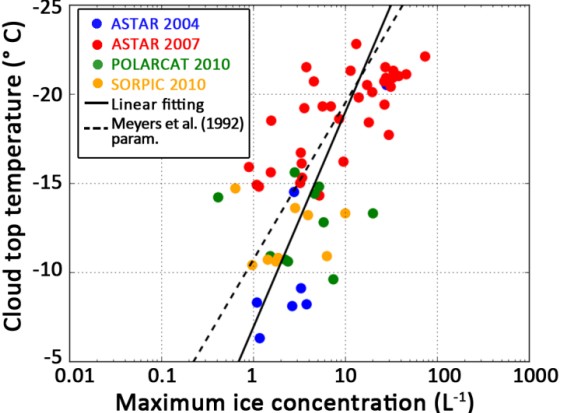



**Figure 10: Maximum ice crystal concentration as a function of cloud top temperature. The colored circles represent**
**the values for each profile (with fitting in dashed line). The Meyers et al. (1992) parameterization is also displayed**
**(dotted line).**








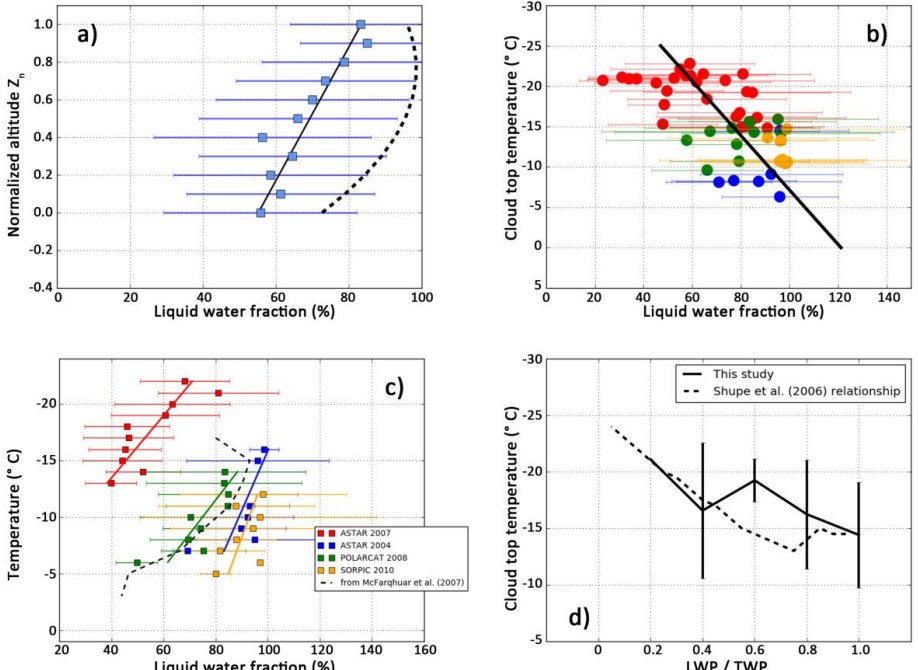

**Figure 11: Liquid water fraction according to $Z_n$ (a), cloud top temperature (b) and temperature (c). The dotted**
**dashed line on panels a) and c) is the parameterization from McFarquhar et al. (2007) and the solid lines on panels a),**
**b) and c) are the fittings for the present study. d) Ratio of LWP over TWP according to cloud top temperature. The**
**solid line refers to the present study and the dotted lines refers to Shupe et al. (2006) work.**


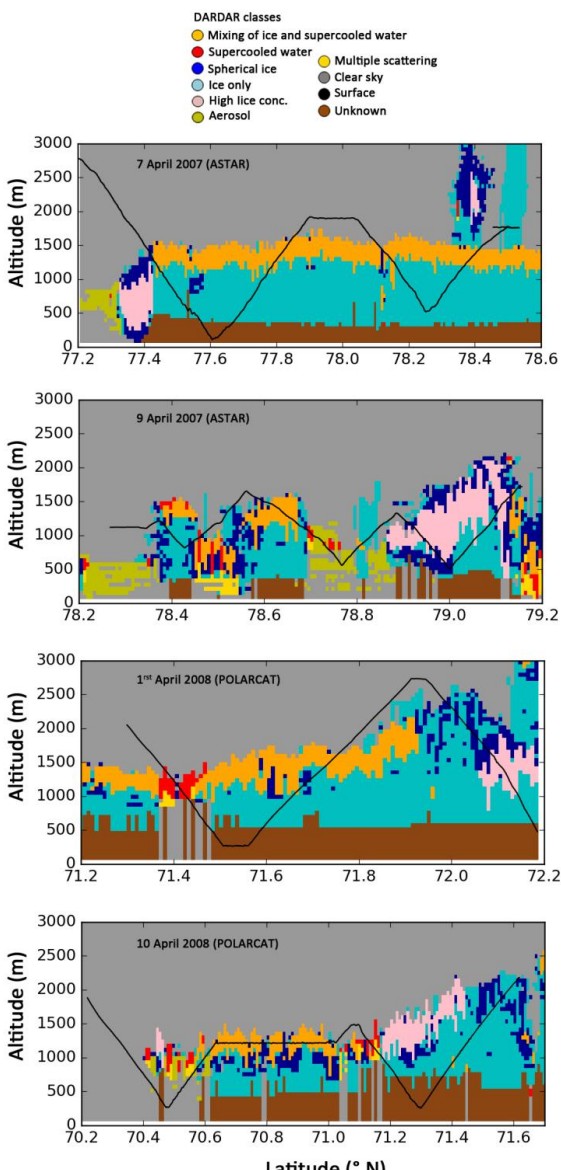

**Figure 12: Vertical profiles of the DARDAR cloud phase for the four satellite validation situations encountered during ASTAR 2007 (7 and 9 April 2007) and POLARCAT 2008 (1srt April and 10 April 2008). The black lines show the aircraft flight track.**












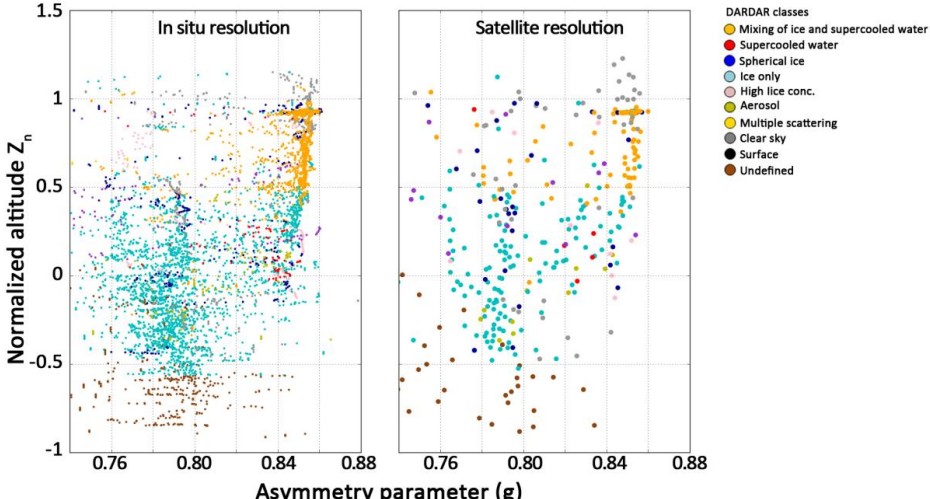

**Figure 13: Vertical profiles of the asymmetry parameter from PN, with the corresponding DARDAR cloud types**
**superimposed in color. Left panel shows the results at the in situ resolution (1Hz ~100m horizontal), right panel at the**
**DARDAR resolution (~1.7km).**





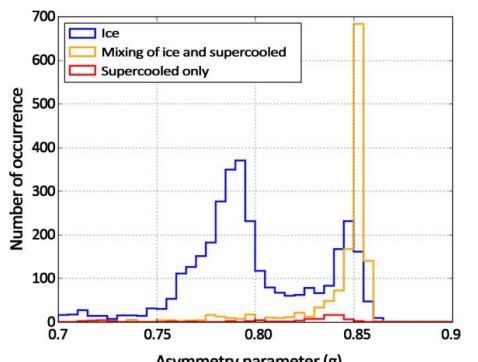 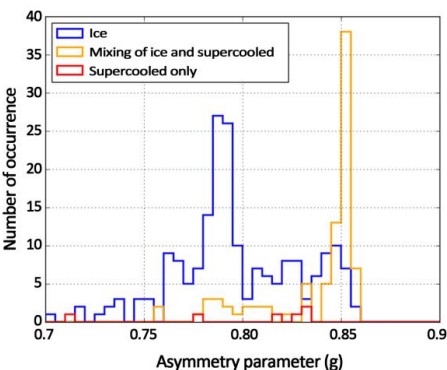



**Figure 14: Frequencies of occurrence of the asymmetry parameter from PN according to the DARDAR cloud phase**
**retrieval in color. Left panel shows the results at the in situ resolution (1Hz ~100m horizontal), right panel at the**
**DARDAR resolution on the right panel (~1.7km).**






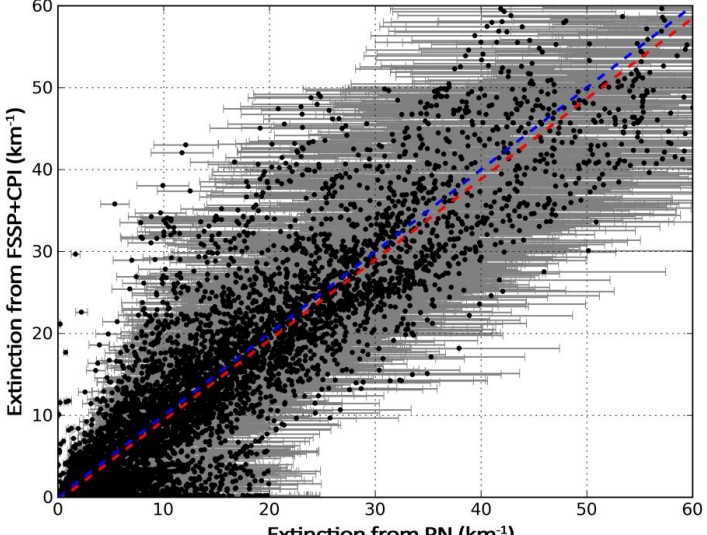



**Figure B1: Comparison of extinction from PN and CPI+FSSP measurements. Grey bars represent the 25 %**
**uncertainties on the PN extinction. The red dotted line is the linear fitting (slope of 0.98, R² = 0.87) and the blue dotted**
**line is the 1:1 line.**