# Peer review of "Characterization of Arctic mixed-phase cloud properties at small scale and coupling with satellite remote sensing."

_Atmospheric Chemistry and Physics, 2017_

## Referee Comment (RC1) · Anonymous Referee #1 · 20 Mar 2017

This is a good paper looking at Arctic mixed phase clouds. Polar clouds in general are poorly studied and this study is a significant contribution to the subject. It looks at in situ data from 18 flights over 4 field campaigns. These campaigns are all during April or May so these flights only look at conditions in spring close to Spitzbergen and although this might seem limiting the paper does show this work can be extended using satellite data. Although I think it should be published and I would not have too many problems if it were published unchanged I have some queries and suggestions that I feel might make it even better – or at least make it easier for me the understand. These are in the order they appear – not in order of importance.

1) (page 5 line 189 and appendix B)I am a little unhappy about the dismissal of the

effect of shattering on the probes. I understand that this is difficult with older probes but surely the CPI would have provided inter arrival time information and possibly visual evidence of shattering – this would then add weight to the comparison of the extinction coefficients in appendix B. Also were the clouds heated – I don't think this is mentioned.

2) (page 5 line 195) Was the air temperature probe heated and what type of probe was used? A heavily iced probe can have a large influence on the air temperature.

3) (Page 6 line 219) Surely there are only 5 normalised levels in the cloud?

4) (Page 8 line 278..) It would be interesting to compare the LWC values in clouds with that obtained with an adiabatic ascent from the cloud base – this would give further confidence in the observed LWC values.

5) (page 9 line 323 ..) It would be interesting to see how the habit (and the number) of the ice crystals observed varied with temperature as well as with height.

6) I am surprized at the lack of needles reported – there appear to be clouds warmer than -10 degC where I would expect the Hallet Mossop process to be important and at these temperatures I would expect there to be columns present ( see Lloyd et all ACP 15 p. 3719-3737)

7) (page 13 line 503 and figure 9b) And why does figure 9b show no ice above -10deg C. I feel that this apparent absence of secondary ice at warmer temperature should be discussed – if only briefly.

8) (page 15 line 579 onwards) I think comparison between the satellite data and the observations is very useful. However I would have liked more description of the DARDAR scheme and how retrievals method used to determine cloud phase differs between the satellite and in situ measurements. Are the two retrieval methods looking at different things.

9) (Page 17 line 655) I cannot really see the pink in figure 13

---

## Referee Comment (RC2) · Anonymous Referee #2 · 3 Apr 2017

**1   General remarks**

The manuscript provides a broad analysis of airborne in situ measurements of microphysical and optical properties of Arctic low-lewel mixed-phase clouds. Data of four field campaigns are merged and presented with respect to different aspects in order to improve our understanding of this specific but most relevant Arctic cloud type. In a second part in situ measurements and satellite observations are compared. This analyze focus on the question if the spatial resolution of satellite to resolve all small scale aspects of mixed-phase clouds or if averaging may bias our view on these clouds.

The large data bases presented and analyzed in the manuscript and the derived pa-

rameterisations are of high value for Arctic climate research. Also the case study on the performance of active satellite remote sensing will help to improve our understanding of the retrieval products and the interpretation of derived cloud properties. In this regard, the manuscript provides an important contribution to current and future research and is worth to be published.

However, in my opinion the manuscript lacks of some major issues which have to be reassessed in detail before publishing the manuscript. Especially the focus of the manuscript is not sufficiently clear to the reader. Two topics are presented but not combined in the manuscript. I therefore recommend to split the manuscript into two single contributions that strongly focus on the individual subjects. This will certainly enhance the visibility of both studies as both will suffer when presented in a single manuscript.

Below, I compiled a list of comments which have to be considered in a revised version of the paper. There might be some contradictory statements which result from the sometimes not well clarified differentiation of the two separate studies. I still kept most of my thoughts in the reviews as they might help to each of the single studies. I also have to admit, that I did not review the second study on the satellite comparison with the same depth as the first part of the manuscript, as I soon had the feeling that both parts should be split.

**2  Major comments**

**Two individual studies**

The manuscript presents two different parts with different application of the measured cloud properties. First all in situ cloud data are systematically analyzed and new parameterizations are presented. In the second part a completely different topic, namely a case study of satellite-airborne comparison is presented.

I do not see a large overlap, connection or synergy of both parts, except that measurements of the same instruments and field campaigns (only two out of four) are used. The approach of both parts is different: statistical analysis of a large data set versus comparison in single case studies. The satellite comparison study does not use the full data set described before and is a completely stand alone study. This is already obvious as the second part starts with an additional and separate introduction trying to justify the step from the statistical analysis to the case studies.

I was also confused by the misleading title of the paper which I guess is more related to the title of the satellite comparison in the second part of the paper. I don't see a characterization on small scales in the first part of the manuscript. Cloud profiles which are averaged over a relatively large area were analysed. Therefore, the title neglects the major part of the manuscript and is misleading.

Overall, I therefore recommend to split both parts of the manuscript and publish separately. The statistical analysis of the large in situ data set seems to be more mature and provides interesting results. That would be my first choice of the focus for this paper.

Nevertheless, I like the idea and the results of the second part, satellite comparison. The two plots show quite nicely how averaging may effect the cloud properties in terms of separating ice and liquid parts of the cloud. Therefore, I recommend to publish this in a separate publication. This would also enhance the visibility of this study. Keeping the comparison study here as a second part following the extended statistical analysis of the whole data set is remains kind of hidden.

**"Small scale"**

Section 3 is called "Small scale properties of liquid droplets and ice crystal particles within MPC" but averaged profiles are analyzed. These profiles can not resolve small scale horizontal variability. Also 100 m vertical resolution is not really "small" scale considering that typical mixed-phase clouds have a quite small vertical extend of a

couple of 100 meters. Remove "small scale" for this section.

The title of the manuscript also refers to "small scale". This can not be attributed to the statistical analysis and only holds for the comparison with the coarser satellite data.

**Focus of the data analysis**

After reading the first section, I feel that the focus of the manuscript was not clearly defined in the beginning. The analysis of the data set looks slightly arbitrary. Some dependencies are analyzed campaign wise, some with the entire data set. Also the discussion or analysis with other parameters that might influence some of the found dependencies short.

To improve the outcome of the analysis, a stronger focus and master plan of how to analyse the data in a consistent way is needed. A more systematic and logic analysis of the data is necessary.

My general suggestion is to use the data set in order to decouple dependencies of different cloud microphysical properties to different environmental parameters. This is what the data set can be used for. Here, the separation between the single campaigns as often done in the manuscript is misleading. Why there are differences between the campaigns? E.g., because of temperatures. But, even when the microphysical properties may differ between the campaigns, the variability between different flights might be larger as the duration of the campaigns is likely longer than the synoptical time scale and conditions will change during a campaign. It is not the campaign name and year what matters it is rather the synoptic conditions and these might differ also within a campaign. Therefore, I suggest to not separate the data into different campaigns. Rather separate in different synoptic situation, e.g., temperature regimes, wind direction, vertical wind speed, etc.

**Aerosol Effect**

The aerosol effect discussed in Section 3.4 is not really convincing. Even the authors rise doubt on their on conclusions. There is a lot of speculations in this paragraph, "*cloud*", "*may*", no airborne aerosol sampling is available and only few cases are used to testify the potential influence of the aerosol concentration. Finally, the authors kill their arguments by themselves in stating, that "*the aerosol concentration measurements have to be taken with care*", and "*Measurements of key parameters are obviously missing in the present study to accurately quantify the mechanisms responsible for the formation and growth of droplets and ice crystals within MPC*".

With this, I only can recommend to remove this highly uncertain and speculative analysis. You may provide the information on the aerosol background for completeness but not try to draw any conclusion on the aerosol effect.

**3   Minor comments**

**L11:** Title: You may add that the study is only on low-level mixed-phase clouds.

**L12:** Title: What is a "small scale"? 10 km or 1 m?

**L18:** Which season the campaigns have been conducted?

**L21:** "The ice phase is found everywhere within the MPC layers..." this is a somehow trivial statement, as a mixed-phase layer is defined by having both liquid and ice particles.

**L58:** Give a range of the scales for typical model grid boxes and the isolated pockets discussed by Korolev and Isaac (2003). This is important to know with respect to your investigation of small scale variability.

**L64:** After this paragraph I would expect a discussion on the different scales of observation. What are the scales of the cloud phase distribution inhomogeneities and what instruments can resolve it. I'm not sure, if this is the correct place, but somewhere this discussion should be added.

**L67:** Again, specify the "regional" scale.

**L81:** What scales are covered by the "microphysical scale"? mm? $\mu$m?

**L89:** "in situ" is sometimes written italic sometimes not. Be consistent.

**L92:** This means you implicitly assume differences in cloud properties for the different regions, western Arctic and Svalbard? State that directly and give a reference.

**L110:** I'm missing a summary of available similar data sets. In situ cloud properties of mixed-phase clouds have been reported since many years (e.g., Fleishauer, Lawson, Korolev, McFarquhar, Fridlind,...). What data is available? What is their limitation? What is missing on the data? And why this new data set is needed? What can it do better?

**L125:** It is a little irritating that you spend three numbers for four campaigns.

**L128:** Does the flight speed of the aircraft used in the different campaigns significantly differ? Would this have any impact on the spatial resolution if the measurements and according results?

**L138:** The areas have a slightly different latitude and therefore distance to sea ice edge. Could this change the cloud properties between the single campaigns?

**L142:** What is the horizontal range needed to sample a single profile? Additionally, you averaged profiles. When merging these data, for what area the measurements are representative then if you consider the horizontal inhomogeneities of mixed-phase clouds as discussed by Korolev and Isaac (2006)? (Korolev, A. and G.A. Isaac, 2006: Relative Humidity in Liquid, Mixed-Phase, and Ice Clouds. J. Atmos. Sci., 63, 2865–2880, doi: 10.1175/JAS3784.1.)

**L165:** Operated with or without Korolev tips? Is shattering a problem?

**L176:** "Liquid Water Content" - no capital letters.

**L192:** Give at least a conclusion of the Appendix here in the main text. I also would suggest to place Table A1 in the main part of the manuscript and not in the appendix. The uncertainty estimates are essential to interpret the measurements and should not be hidden in an Appendix.

**L210:** I was totally lost because the equations are not consistent. Until I realized that the equations are used for different altitudes. Indicate here for which range of $z$ ($z > zb$ and $z < zb$) the equations are applied.

**L220:** How you can guaranty the number of 2000 observations? Can you tell a little about flight strategy? Continuous profiles or stair cases? What ascent/descent rates were flown?

**L221:** Typo: "ofbtained"

**L229:** remove "very". That's always relative.

**L231:** The numbers given in Table 2 are not the normalized altitudes, right? This is a little confusing, as you first introduced the normalized altitudes and the temperature profiles and then use the geometric altitudes again. I would recommend to reorder this.

**L231:** If the temperatures did differ that strongly, do the cloud base and top altitudes also vary between the single campaigns? I would also split this up and somehow combine the numbers with Figure 2 instead of given the numbers in Table 2. That would be more illustrative.

**L232:** Well, for cloud base the observations and literature are not consistent anymore. The mean value of your study is off the range observed by McFarquhar et al. (2007).

**L246:** Is there a way to give a number from which fraction of IWC/TWC the influence of ice might be visible in the scattering data?

**L250:** These two thresholds of g differ. Do I understand correctly, that for the range 0.8-0.83 both, liquid and ice crystal properties were derived? Please add this shortly to clarify to the reader.

**L253:** I don't think, that Table 3 is needed, when you properly describe the matrix in the text here. Except of the range g=0.8-0.83 everything was given in the text.

**L254:** I would suggest to place this essential sentence a bit earlier. Maybe at line 248.

**L265-267:** You don't have to repeat this. Remove sentence.

**L276:** "droplet number size distribution" is the correct name of the quantity.

**L297:** Here only particle larger 100 $\mu$m are considered. But all other plots and analyzed parameters include particles less than 100 $\mu$m, e.g., number size distributions and $D_{\text{eff}}$. Be consistent or give a reasons why not to be consistent.

**L305:** Introduce acronyms ISDAC and MPACE.

**L305:** A more detailed discussion with this available data would be helpful.

**L319:** In this analysis the data of the four campaigns are merged. Is there evidence for differences in the particle shapes considering the strong temperature difference during ASTAR 2007?

**L337:** add "at low altitudes" for "...MPC in the Svalbard region at low latitudes."

**L345:** This section analyses phase function and $g$ and introduced these quantities. However, $g$ values were already extensively used in the sections before to discriminate the cloud particle phase. The presentation of $g$ and discussion has to be given first.

By presenting $g$ and phase function at this position of the manuscript, the analysis does not give any further new results and insight. The general location of ice and liquid particles was already discussed before. I therefore, recommend to shift this discussion somewhere earlier, before the first $g$ values are used.

**L374:** I would also consider the sampling method to cause a mixing or averaging at cloud top. Assuming a cloud with variable cloud top altitude (even if only a couple of meters) and the aircraft ascending/descending through cloud top with a certain horizontal speed. If the ascending/descending rate is to low, then different parts of the horizontal inhomogeneous cloud top are averaged. This automatically will lead to lower LWC when averaging cloudy and cloud free patches.

**L376:** Typo: "an" ice crystal growth...

**L382:** What about vertical wind speed? Was this measured and can this be included in the analysis. Especially for such microphysical processes, the updraft velocity is important (Korolev, A. and P.R. Field, 2008: The Effect of Dynamics on Mixed-Phase Clouds: Theoretical Considerations. J. Atmos. Sci., 65, 66–86, doi: 10.1175/2007JAS2355.1.)

**L383:** LWC: Shortly explain how this was calculated. On what measured data the calculation is based?

**L392:** In what there is coherence with previous work? Such unspecific statements occur more often throughout the manuscript. It is always mentioned, that there is agreement, but it is not discussed in detail, what i particular agrees. Please look though the manuscript and be more specific.

**L466:** This is a repetition. It was just written before.

remove the whole introduction of 4.

from a modeling perspective also other parameter are needed to be correlated with the cloud properties. But these are not adressed here because no measurements are available. So I would not start with this motivation...

**L4:**

**L4:**

**L484:** How does this compare to standard models of the relationships between optical

thickness and LWP. Actually these standard model equations include the particle size, which was not done here.

**L489:** The upper range is missing for the temperature range of this study.

**L492:** "very few previous studies": There are plenty of studies analyzing the properties in Arctic MPC although many use ground based remote sensing (Shupe, De Boer, etc). Some of them are also given in the test below.

**L503:** What is the hypothesis in linking cloud top temperature and IWP, LWP?

**L516-525:** This paragraph is rather part for the motivation. Here in the main text results should be presented and discussed. Add this discussion (state of the art microphysical processes...) to the introduction.

**L529:** Give the uncertainties for the fit parameter. There are statistical means to calculate these. Only if the uncertainty of the fit parameter covers the parametrization by Meyers et al. (1992) you can judge if both data sets agree with each other.

**L531:** Remove the last sentence. This is more useful for the conclusions and outlook.

**L537:** rewrite: "...liquid fraction of individual measurements can be...". The liquid fraction of an entire MPC would be LWP/TWP. Differentiate both parameters of liquid fraction for indivudual measurements and for the entire cloud. Spend symbols for both and introduce carefully.

**L570:** That first has to be shown! Especially with the good agrement with available parametrization, what is in particular the benefit of the new data set?

**L571-575:** This fits better into the motivation. Move to introduction.

**L760:** Conclusions: Of course it is a question of style. But if you have such long conclusions I see no benefit in putting these into a numeration. It could also be presented by regular paragraphs.

**L762-765:** The first two sentences are not a conclusion. Try to shorten and focus your conclusions.

**L792:** This first has to be shown. You only presented the parameterizations but did not apply to remote sensing nor modelling.

**L802:** "Globally"? You just looked at 4 flights.

**L1030-1033:** Reference is duplicated.

**Figure 1:** It would help if you indicate where the single profiles have been collected. The total flight paths are more or less irrelevant as they do not show where the profiles are measured.

**Figure 3 and 4:** Figures and labels are way to small! Even zooming in to 400%, I can not read the label. Spend more space for these figures. Maybe a 2x2 arrangement would help.

**Figure 5:** The liquid droplets shown here are measurements from the FSSP? Or do you found liquid droplets larger than 100 $\mu$m with the CPI?

**Figure 5:** Between 0 and -3°C almost only liquid droplets are found. Is this liquid also the liquid droplets at cloud top in the upper panels? May be only measured above the inversions? Or is this smaller fraction of liquid and cloud top compared to high temperatures only due to bad statistics?

**Figure 6:** This plot shows an average over all campaigns. Are there differences between the campaigns? I would assume similar differences compared ot the LWC and IWC profiles.

**Figure 9:** I'm not sure, if the log-scales somehow automatically gives a "good" fit. In linear space larger differences might be visible.

**Figure 9:** Typo: Write "LWC" for the lower equation, liquid droplets.

**Figure 9:** x-title: write "extinction coefficient"

**Figure 10:** In Fig. 8 cloud top temperature was given on the x-axis, here on y-axis. Make this consistent.

**Figure 11:** Panel a: Here the campaigns are not separately displayed. Why?

**Figure 11:** Panel b and c: How there can be error bars beyond 100%?

[Figure]

---

## Author Comment (AC1) · 29 Jul 2017

**Vertical distribution of microphysical properties of Arctic springtime low-level mixed-phase clouds over the Greenland and Norwegian Seas.**

(initial title was: **Characterization of Arctic mixed-phase cloud properties at 1 small scale and coupling with satellite remote sensing**.)

**By G. Mioche et al.**

The authors would like to thank the reviewer for his/her helpful comments and suggestions, which hopefully will help us to greatly improve the quality of our paper.
Below you will find detailed answers to the reviewer's comments:
- **the reviewer's comments (RC) are reported in bold**,
- *the authors' responses (AR) are reported in italic,*
- *the changes in the revised manuscript are indicated in italic and red color.*

All the manuscript has been rewritten, title has been modified and numerous additions and changes concerning the structure have been made in order to take into account the reviewer's comments. For this reason, the whole additional text included in the revised manuscript could not be indicated in this document.

In regards to the major comments made by reviewer 2 and to keep focus in the paper, the part dealing with satellite remote sensing validation (section 5, figures 12, 13 and 14 and tables 4, 5, 6, and 7) has been removed and will be the topic of a future paper.
* * *
**1)**
**(page 5 line 189 and appendix B)**
**Referee comment (RC): I am a little unhappy about the dismissal of the effect of shattering on the probes. I understand that this is difficult with older probes but surely the CPI would have provided inter arrival time information and possibly visual evidence of shattering – this would then add weight to the comparison of the extinction coefficients in appendix B. Also were the clouds heated – I don't think this is mentioned.**
*Authors' response (AR): The CPI used is the version 1. It has a maximum image capacity of 40 images per seconds, thus the maximum time resolution between two images (i.e. the maximum inter-arrival time resolution) is 25ms. Measurements from **Korolev et al. (2013)** made with the NRC Convair-580 in similar flight conditions show on figure 14 that the inter-arrival times of shattered particles are below 1 ms. The CPI cannot produce evidence of shattering thanks to the particle arrival time. Visual evidence of shattering is difficult (at best) has shown by **Schwarzenboeck et al. (2009)**.*
*All the cloud probes were heated to avoid icing during the flights.*

**2)**
**(page 5 line 195)**
**RC: Was the air temperature probe heated and what type of probe was used? A heavily iced probe can have a large influence on the air temperature.**
*AR: The types of probes used are:*
- *Rosemount sensor during the POLARCAT and ASTAR campaigns,*

- *and the Advanced Airborne MeasureMent Solutions system (AIMMS-20) during the SORPIC campaign.*

*All the air temperature probes were heated to avoid icing during the flights. This has been added line 204.*

**3)**
**(Page 6 line 219)**
**RC: Surely there are only 5 normalised levels in the cloud?**
*AR: We used 5 normalized levels to have statistically representative levels. More than 5 levels in the cloud layer can lead to some level with a very weak number of observations and thus averaged profiles not really representative.*

**4)**
**(Page 8 line 278.)**
**RC: It would be interesting to compare the LWC values in clouds with that obtained with an adiabatic ascent from the cloud base – this would give further confidence in the observed LWC values.**
*AR: It has been done based on the calculation of the ratio of observed LWC values on the adiabatic LWC values in figure 7 and discussed in the text. It gives an indication on the efficiency of ice processes (WBF, riming), as well as evaporation of liquid droplets at cloud top.*

**5)**
**(page 9 line 323 ..)**
**RC: It would be interesting to see how the habit (and the number) of the ice crystals observed varied with temperature as well as with height.**
*AR: It is done in the panels c) and d) in Figure 5 (now Figure 6 in the revised manuscript).*

**6)**
**RC: I am surprized at the lack of needles reported – there appear to be clouds warmer than -10 degC where I would expect the Hallet Mossop process to be important and at these temperatures I would expect there to be columns present (see Lloyd et all ACP15 p. 3719-3737)**
*AR: Ice shape depends on the temperature when ice nucleation occurs, and this temperature can be different of that of in situ observations. This could explained that needles are not observed above -10°C. Moreover, there are few data corresponding to temperatures warmer than -10°C.*

**7)**
**(page 13 line 503 and figure 9b)**
**RC: And why does figure 9b show no ice above -10deg C. I feel that this apparent absence of secondary ice at warmer temperature should be discussed – if only briefly.**
*AR: IWP is very small above -10°C, but is not equal to 0. Moreover, Figure 9b (now Figure 10b in the revised manuscript) displays IWP according to the cloud top temperature (i.e. the coldest temperature for each profile) and very few profiles have a cloud top temperature above -10°C.*

**8)**
**(page 15 line 579 onwards)**
**RC: I think comparison between the satellite data and the observations is very useful. However I would have liked more description of the DARDAR scheme and how retrievals method used to determine cloud phase differs between the satellite and in situ measurements. Are the two retrieval methods looking at different things.**

*AR: Detailed description of DARDAR scheme is made in **Ceccaldi et al. (2013)**, and **Delanoë and Hogan (2008, 2010)**, but section 5 has been removed and will be the topic of a separate paper.*

**9)**
(Page 17 line 655)
**RC: I cannot really see the pink in figure 13**
*AR: Figure 13 has been removed since section 5 has been removed.*

**References**

Ceccaldi, M., Delanoë, J., Hogan, R. J., Pounder, N. L., Protat, A. and Pelon, J.: From CloudSat-CALIPSO to EarthCare: Evolution of the DARDAR cloud classification and its comparison to airborne radar-lidar observations, J. Geophys. Res. Atmospheres, 118, 1–20, doi:10.1002/jgrd.50579, 2013.

Delanoë, J. and Hogan, R. J.: A variational scheme for retrieving ice cloud properties from combined radar, lidar, and infrared radiometer, J. Geophys. Res., 113(D07204), doi:10.1029/2007JD009000, 2008.

Delanoë, J. and Hogan, R. J.: Combined CloudSat-CALIPSO-MODIS retrievals of the properties of ice clouds, J. Geophys. Res., 115(D0029), doi:10.1029/2009JD012346, 2010.

Korolev, A. V., Emery, E. F., Strapp, J. W., Cober, S. G. and Isaac, G. A.: Quantification of the Effects of Shattering on Airborne Ice Particle Measurements, J. Atmospheric Ocean. Technol., 30(11), 2527–2553, doi:10.1175/JTECH-D-13-00115.1, 2013.

Schwarzenboeck, A., Shcherbakov, V., Lefevre, R., Gayet, J.-F., Pointin, Y. and Duroure, C.: Indications for stellar-crystal fragmentation in Arctic clouds, Atmospheric Res., 92(2), 220–228, doi:10.1016/j.atmosres.2008.10.002, 2009.

---

## Author Comment (AC2) · 29 Jul 2017

**Vertical distribution of microphysical properties of Arctic springtime low-level mixed-phase clouds over the Greenland and Norwegian Seas.**

(initial title was: **Characterization of Arctic mixed-phase cloud properties at 1 small scale and coupling with satellite remote sensing**.)

**By G. Mioche et al.**

The authors would like to thank the reviewer for his/her helpful comments and suggestions, which hopefully will help us to greatly improve the quality of our paper.
Below you will find detailed answers to the reviewer's comments:
- **the reviewer's comments (RC) are reported in bold**,
- *the authors' responses (AR) are reported in italic,*
- *the changes in the revised manuscript are indicated in italic and red color*.

All the manuscript has been rewritten, title has been modified and numerous additions and changes concerning the structure have been made in order to take into account all the reviewer's comments. For this reason, the whole additional text included in the revised manuscript could not be indicated in this document.
* * *
**Major comments**

- **Two individual studies**

**Referee Comment (RC): The manuscript presents two different parts with different application of the measured cloud properties. First all in situ cloud data are systematically analyzed and new parameterizations are presented. In the second part a completely different topic, namely a case study of satellite-airborne comparison is presented.**

**I do not see a large overlap, connection or synergy of both parts, except that measurements of the same instruments and field campaigns (only two out of four) are used. The approach of both parts is different: statistical analysis of a large data set versus comparison in single case studies. The satellite comparison study does not use the full data set described before and is a completely stand alone study. This is already obvious as the second part starts with an additional and separate introduction trying to justify the step from the statistical analysis to the case studies.**

**I was also confused by the misleading title of the paper which I guess is more related to the title of the satellite comparison in the second part of the paper. I don't see a characterization on small scales in the first part of the manuscript. Cloud profiles which are averaged over a relatively large area were analyzed. Therefore, the title neglects the major part of the manuscript and is misleading.**

**Overall, I therefore recommend to split both parts of the manuscript and publish separately.**
**The statistical analysis of the large in situ data set seems to be more mature and provides interesting results. That would be my first choice of the focus for this paper.**

**Nevertheless, I like the idea and the results of the second part, satellite comparison. The two plots show quite nicely how averaging may affect the cloud properties in terms of separating ice and liquid parts of the cloud. Therefore, I recommend to publish this in a separate publication. This would also enhance the visibility of this study. Keeping the comparison study here as a second part following the extended statistical analysis of the whole data set is remains kind of hidden.**

*Authors' response (AR): Section 5 focusing on satellite remote sensing validation has been removed from the manuscript according to the reviewer's recommendation. It will be the topic of a separate paper. The revised manuscript focuses now only on liquid droplets and ice crystal microphysical properties of MPC and the title has been changed to:*
*"Vertical distribution of microphysical properties of Arctic springtime low-level mixed-phase clouds over the Greenland and Norwegian Seas"*

- **"Small scale"**

**RC: Section 3 is called "Small scale properties of liquid droplets and ice crystal particles within MPC" but averaged profiles are analyzed. These profiles can not resolve small scale horizontal variability. Also 100m vertical resolution is not really "small" scale considering that typical mixed-phase clouds have a quite small vertical extend of a couple of 100 meters. Remove "small scale" for this section.**

*AR: The title of this section has been change to "Vertical properties of liquid droplets and ice crystal particles within MPCs"*

**RC: The title of the manuscript also refers to "small scale". This can not be attributed to the statistical analysis and only holds for the comparison with the coarser satellite data.**

*AR: The title of the manuscript is now: "Vertical distribution of microphysical properties of Arctic springtime low-level mixed-phase clouds over the Greenland and Norwegian Seas"*

- **Focus of the data analysis**

**RC: After reading the first section, I feel that the focus of the manuscript was not clearly defined in the beginning. The analysis of the data set looks slightly arbitrary. Some dependencies are analyzed campaign wise, some with the entire data set. Also the discussion or analysis with other parameters that might influence some of the found dependencies short.**

**To improve the outcome of the analysis, a stronger focus and master plan of how to analyse the data in a consistent way is needed. A more systematic and logic analysis of the data is necessary.**

**My general suggestion is to use the data set in order to decouple dependencies of different cloud microphysical properties to different environmental parameters. This is what the data set can be used for. Here, the separation between the single campaigns as often done in the manuscript is misleading. Why there are differences between the campaigns? E.g., because of temperatures. But, even when the microphysical properties may differ between the campaigns, the variability between different flights might be larger as the duration of the campaigns is likely longer than the synoptical time scale and conditions will change during a campaign. It is not the campaign name and year what matters it is rather the synoptic conditions and these might differ also within a campaign. Therefore, I suggest to not separate the data into different campaigns. Rather separate in different synoptic situation, e.g., temperature regimes, wind direction, vertical wind speed, etc.**

*AR: We agree with the reviewer that a classification based on the campaign may not be that relevant. The original idea was that a large temperature range exists between the campaigns, in particular*

between the ASTAR 2007 (cold situations) and the other campaigns which were performed in warmer situations. That is why we decided to show the averaged profiles for each campaign. Moreover, all the selected situations correspond to a same cloud type (low-level boundary layer mixed-phase clouds).

However the reviewer is right: the meteorological situation can vary from one campaign to another which implies that a better "synoptical" classification is needed. The goal of this paper is to provide a statistical study of MPC observations stemming from several campaigns. So, it does not seem reasonable to undertake a detailed description of each meteorological situation. However, we have carried out a classification based on the cloud top temperature and the air mass origin in order to separate the dataset according to the environmental conditions instead of the experiment timeline :

- Two temperature regimes have been selected according to the mean cloud top temperature of each situation: the "cold" situations (-22 °C < $T_{Top}$ < -15°C) and the "warm" situations (-15 °C < $T_{Top}$ < -8°C).

- Air mass origins have been investigated based on the HYSPLIT back-trajectories and two categories have been chosen:  air masses coming from the North (Arctic Ocean) and the air masses coming from the South and/or East (more continental regions).  The dataset has been divided in only 4 classes (2 temperature regimes and 2 air mass origins) in order to ensure the statistical significance of each class.

*Table 1 summarizes these different regimes and has been added to the manuscript.*
One can note that all the cold situations are correlated with a northern origin of the air mass (blue in Table 1). Among the 12 warm situations, 7 correspond to air masses from North (green in Table 1) and 5 from South/East (red in Table 1). Based on these temperatures regimes and air mass origins, three distinct situations can be discriminated : The cold ones (with air mass coming from North), hereafter referred as "COLD" in the manuscript, the warmer situations with air masses from North (hereafter WARM_NO) and warmer situations with air masses from South (i.e. continental, hereafter WARM_SO)
*This classification has been described in a new section (2.4. Meteorological situations) and all the manuscript (vertical profiles, discussions and parameterizations) and figures (Figs. 2, 3, 4, 7, 10 and 11) have been revised according to this new classification.*

| Experiment | Date | Mean $T_{Top}$ (°C) | Air mass origin N = North (Arctic Ocean) S/E = South or East (Continental | Regime |
|---|---|---|---|---|
| ASTAR 2004 | 15 may | -16,5 | N | COLD |
| | 22 may | -8,5 | S | WARM_SO |
| | 25 may | -8 | N | WARM_NO |
| | 5 june | -11 | N | WARM_NO |
| ASTAR 2007 | 2 april | -21 | N | COLD |
| | 3 april | -16 | N | COLD |
| | 7 april | -22 | N | COLD |
| | 8 april | -19 | N | COLD |
| | 9 april | -21 | N | COLD |
| POLARCAT 2008 | 31 march | -15 | N | WARM_NO |
| | 1$^{srt}$ april | -10 | N | WARM_NO |
| | 10 april | -14 | N | WARM_NO |
| | 11 april | -14 | N | WARM_NO |
| SORPIC | 4 may | -13 | SE | WARM_SO |

| | | | | |
|---|---|---|---|---|
| 2010 | 5 may | -11 | SE | WARM_SO |
| | 6 may | -13 | SE | WARM_SO |
| | 9 may | -15 | S | WARM_SO |
| | 10 may | -13,5 | N | WARM_NO |

**Table 1: Classification of the MPC situations according to temperature regimes and air mass origins**

- **Aerosol Effect**

**RC: The aerosol effect discussed in Section 3.4 is not really convincing. Even the authors rise doubt on their on conclusions. There is a lot of speculations in this paragraph, "cloud", "may", no airborne aerosol sampling is available and only few cases are used to testify the potential influence of the aerosol concentration. Finally, the authors kill their arguments by themselves in stating, that *"the aerosol concentration measurements have to be taken with care",* and *"Measurements of key parameters are obviously missing in the present study to accurately quantify the mechanisms responsible for the formation and growth of droplets and ice crystals within MPC".***

**With this, I only can recommend to remove this highly uncertain and speculative analysis.**
**You may provide the information on the aerosol background for completeness but**
**not try to draw any conclusion on the aerosol effect.**

*AR: This analysis has been significantly reduced. However, the values of aerosol concentration measured at the Zeppelin station are still mentioned in the manuscript since it still provides an insight of the aerosol loading which is in addition consistent with the air mass origin classification.*

**Minor comments**

**RC: L11: Title: You may add that the study is only on low-level mixed-phase clouds.**
*AR: done*

**RC: L12: Title: What is a "small scale"? 10km or 1m?**
*AR: It refers to around 100 m. But since the satellite section has been removed, the term "small scale" has been removed too.*

**RC: L18: Which season the campaigns have been conducted?**
*AR: The campaigns were conducted during the spring season. This has been added in the abstract (line 17) and title.*

**RC: L21: "The ice phase is found everywhere within the MPC layers..." this is a somehow trivial statement, as a mixed-phase layer is defined by having both liquid and ice particles.**
*AR: This statement has been removed and the sentence is now: "The ice phase dominates the properties of the MPCs in the lower part of the cloud and beneath it…" (line 22)*

**RC: L58: Give a range of the scales for typical model grid boxes and the isolated pockets discussed by Korolev and Isaac (2003). This is important to know with respect to your investigation of small scale variability.**
*AR: Typical scales of model grid boxes is 100 km horizontal x 1 km vertical. Isolated pockets of liquid and ice are on the order of tens of meters (also in* **Rangno and Hobbs, 2001***)*
*This has been added to the manuscript (lines 73 and 76)*

**RC: L64: After this paragraph I would expect a discussion on the different scales of observation. What are the scales of the cloud phase distribution inhomogeneities and what instruments can resolve it. I'm not sure, if this is the correct place, but somewhere this discussion should be added.**

*AR: We did our best to modify the introduction in order to take this comment into account (lines 64 to 143)*

**RC: L67: Again, specify the "regional" scale.**

AR: "regional scale" has been replaced by: "from a few km to the pan-arctic region" *(line 107)*

**RC: L81: What scales are covered by the "microphysical scale"? mm? _m?**

*AR: We consider here the "microphysical scale" the scale of the in situ measurements i.e. the detection of particles of a few μm, and a spatial resolution typically around 100 m (according to the aircraft speed).*

*The following text has been included: "(i.e. measurements of microphysical cloud properties, spatial resolution less or equal to 100 m)" (line 114)*

**RC: L89: "in situ" is sometimes written italic sometimes not. Be consistent.**

*AR: Fixed*

**RC: L92: This means you implicitly assume differences in cloud properties for the different regions, western Arctic and Svalbard? State that directly and give a reference.**

*AR: Studies of Arctic MPC at the regional scale (such as **Mioche et al., (2015)**) showed that MPC occurrence presents significant seasonal and spatial variability according to the location and the associated specific environmental conditions. For example, the vicinity of Atlantic Ocean may explain in part the large presence of MPC in the Svalbard / Greenland Sea regions all along the year. Also, the sea ice melting may play a role in the variability of MPC occurrence in the Western Arctic regions (Beaufort sea) by the transport of warm water and humidity through the Arctic Ocean from spring to autumn. Moreover, statistical studies of MPC properties in the Western regions have already been conducted (**McFarquhar et al., 2007**), whereas no similar work has been done in the Atlantic side (only cases studies). So, it appears important to investigate the microphysical properties of MPC in the Svalbard / Greenland Sea regions from a statistical point of view to provide representative profiles and then compare to previous works (which has not been done yet).*
*This has been added to the manuscript (lines 123 to 129).*

**RC: L110: I'm missing a summary of available similar data sets. In situ cloud properties of mixed-phase clouds have been reported since many years (e.g., Fleishauer, Lawson, Korolev, McFarquhar, Fridlind,...). What data is available? What is their limitation? What is missing on the data? And why this new data set is needed? What can it do better?**

*AR: Yes, lots of studies have been done with in situ data sets, but it concerns case studies or in the western regions. No work on MPCs in the Greenland Sea regions using several in situ datasets has been done so far. The data presented here are not better than the previous mentioned, the instrumentation is similar to that used for these previous works. The great advantage of the present study is that the same probes have been used for the 4 airborne campaigns, so the determination of cloud properties and the statistical analysis based on the merging of these 4 datasets is very consistent.*

*Moreover, a more consistent comparison with these previous works has been included in section 3 (cf. further comment)*

**RC: L125: It is a little irritating that you spend three numbers for four campaigns.**

*AR: Fixed, we hope it is less irritating now.*

**RC: L128: Does the flight speed of the aircraft used in the different campaigns significantly differ? Would this have any impact on the spatial resolution if the measurements and according results?**

*AR: The typical speed of the aircrafts used in the different campaigns does not significantly differ: it is around 80-100 m/s. So we assume that it does not impact the spatial resolution of the measurements and the results of the present study.*

**RC: L138: The areas have a slightly different latitude and therefore distance to sea ice edge. Could this change the cloud properties between the single campaigns?**

*AR: The distance to the sea ice edge may have an impact on the cloud properties. As shown in **Young et al. (2016)** in the ACCACIA campaign, the transition from sea ice to open water leads to an increase in the cloud base height and the cloud thickness. A small decrease in the droplet number, an increase in their size and LWC values have also been observed, leading to more efficient precipitations over ocean than sea ice surfaces. However, ice properties do not exhibit significant change according to the surface type.*

*In our study, the selected situations concern only MPCs over open water. However, the POLARCAT campaign is located at lower latitudes (around 72°N) compared to ASTAR 2004, 2007 and SORPIC 2010 (around 78°N). The mean cloud thickness during PO08 is 540 m and is slightly greater than the other campaigns (between 290 and 476 m). Moreover, the droplet properties show smaller number of droplets during PO08 than AS04, AS07 and SO10. These findings are consistent with the results from **Young et al. (2016)** and could be attributed to the influence of the distance to sea ice edge.*

*On the other hand, no significant differences on $Z_{base}$ have been observed in the PO08 dataset compared to the other, and droplet size and LWC do not vary according to the distance to sea ice as it is the case in **Young et al. (2016)**.*

*So, clear conclusions on the influence on the distance to sea ice edge cannot been drawn from the present dataset and more in situ data would be necessary. The influence of the surface type and the distance to the ice edge is one of the goals of the ACLOUD campaign which is currently going on at Longyearbyen (Spitzbergen).*

**RC: L142: What is the horizontal range needed to sample a single profile? Additionally, you averaged profiles. When merging these data, for what area the measurements are representative then if you consider the horizontal inhomogeneities of mixed-phase clouds as discussed by Korolev and Isaac (2006)? (Korolev, A. and G.A. Isaac, 2006: Relative Humidity in Liquid, Mixed-Phase, and Ice Clouds. J. Atmos. Sci., 63, 2865– 2880, doi: 10.1175/JAS3784.1.)**

*AR: According to **Korolev et al. (2003)** and **Field et al. (2004)**, the horizontal characteristic scale of phase inhomogeneities is of the order of 100-1000 m. **Korolev and Isaac (2006)** showed that at the horizontal scale of 100 m (or smaller), ice crystals and liquid droplets in a MPC are well mixed, and no isolated pockets of ice or liquid still exist.*

*In our study, clouds are first defined at the 1s time resolution, i.e. 100 m spatial scale, before being averaged. Then, the horizontal range needed to sample a profile is approximately 25 km according to the aircraft speed. Once averaged, each altitude level corresponds to a horizontal range of approximately 2.5 km. Thus, at this horizontal scale, and according to the findings of **Korolev et al. (2003), Field et al. (2004)**, and **Korolev and Isaac (2006)**, our work may be representative of the horizontal inhomogenetities of the cloud phase. However,* the present study focuses more on vertical variability of ice and liquid since only descent and ascent cloud sequences are used. The accurate study of horizontal inhomogeneities should use horizontal flight legs in further studies.

**RC: L165: Operated with or without Korolev tips? Is shattering a problem?**

AR: The FSSP is operated without Korolev tips. So, the shattering could be a problem, that's why we compared the data with CPI and PN measurements (which have different inlet design) to see if there is shattering effect or not.

**RC: L176: "Liquid Water Content" - no capital letters.**

*AR: Fixed*

**RC: L192: Give at least a conclusion of the Appendix here in the main text. I also would suggest to place Table A1 in the main part of the manuscript and not in the appendix.**
**The uncertainty estimates are essential to interpret the measurements and should not be hidden in an Appendix.**
*AR: Done. Table A1 is now Table 2, and the conclusion of the appendix is made at lines 217-220.*

**RC: L210: I was totally lost because the equations are not consistent. Until I realized that the equations are used for different altitudes. Indicate here for which range of z (z > zb and z < zb) the equations are applied.**
*AR: Done at lines 243 and 245.*

**RC: L220: How you can guaranty the number of 2000 observations? Can you tell a little about flight strategy? Continuous profiles or stair cases? What ascent/descent rates were flown?**
*AR: In order to provide consistent vertical profiles of cloud properties, the selected dataset consist only of continuous ascending and descending profiles into clouds (no stair cases) at the aircraft speed around 80-100 m/s. This whole dataset correspond to approximately 21 000 measurement points (as mentioned in section 2.1). Then, the processing of this dataset consists in dividing the cloud layers into 10 normalized levels, each one containing around 2000 measurements points.*

**RC: L221: Typo: "ofbtained"**
*AR: Fixed*

**RC: L229: remove "very". That's always relative.**
*AR: Done*

**RC: L231: The numbers given in Table 2 are not the normalized altitudes, right? This is a little confusing, as you first introduced the normalized altitudes and the temperature profiles and then use the geometric altitudes again. I would recommend to reorder this.**
*AR: The section 2.3 has been reordered:*
*The part dealing with the altitudes in meters (including the old Table 2 (now table 3)) has been moved at the beginning of the section.*
*The part about scattering phase function (old section 3.3) has been added to the section 2.3 (cf. further comment)*

**RC: L231: If the temperatures did differ that strongly, do the cloud base and top altitudes also vary between the single campaigns? I would also split this up and somehow combine the numbers with Figure 2 instead of given the numbers in Table 2. That would be more illustrative.**
*AR: Cloud top and base altitudes slightly vary from one regime to another. The mean cloud top altitude is 1150 m, 1210 m and 1320 m and cloud base altitude is 680 m, 760 m, 930 m for the COLD, WARM_NO and WARM_SO situations respectively. Thus, it appears that clouds are higher for the warmer situations and for a South/East air mass origin.*
*These values have been included with the temperature profiles in the Figure 3 (old Figure 2) and not in Table 3 (old table 2) because the definition of the 3 regimes has to be given first in the manuscript. Table 3 (old Table 2) has been kept in order to provide more detailed information about top and based altitudes (standard deviations, percentiles…), but it is discussed now at the beginning of the section 2.3 (cf. earlier comment)*

**RC: L232: Well, for cloud base the observations and literature are not consistent anymore.**

**The mean value of your study is off the range observed by McFarquhar et al. (2007).**

*AR: In* **McFarquhar et al. (2007)** *cloud base is between 420 and 745 m, so t*he mean value of our study (756 m) is very close to the maximum value of* **McFarquhar et al. (2007).** *Moreover, in our study, the standard deviation is 283 m, so our value may be considered between 473 m and 1039 m. Hence, our altitude range is larger than that of* **McFarquhar et al. (2007),** *but there is an overlap. Furthermore, the determination of cloud base height in* **McFarquhar et al. (2007)** *is based on ground based lidar at Oliktok Point or Barrow (i.e. not directly at the location of in situ measurements) whereas we used in situ PN data to in our study.*

*Moreover,* **McFarquhar et al. (2007)** *also mentioned that the cloud base altitude directly determined from FSSP measurements (and thus more consistent with the technique used in our study) for the 10 October situation is 900 m, which is in the range of our study.*

**RC: L246: Is there a way to give a number from which fraction of IWC/TWC the influence of ice might be visible in the scattering data?**

*AR: In* **Jourdan et al. (2010)***, we have investigated the link between the microphysical properties and the light scattering characteristics of a mixed phase Nimbostratrus cloud during the ASTAR 2004 campaign. This work was done using a principal component analysis on the PN phase function. It was shown (figure 5, p7) that large differences in the side scattering behavior of the phase function were found when the liquid water fraction (LWC/(LWC+IWC))derived from the CPI and Nevzorov probe varied from 100% to 80%. Some substantial differences were also observed for smaller variation of the LWF but were more likely attributed to changes of the particle shape and size (although we could not exclude the contribution of undetected small droplets). For instance, a population composed of 86% droplets, 8% graupels and 5% irregulars (derived from the CPI classification and sorted by aera) had an optical signature different than a population of 100% droplets.  So it is difficult to give specific numbers in terms of IWC/TWC without using an inversion scheme.*

**RC: L250: These two thresholds of g differ. Do I understand correctly, that for the range 0.8-0.83 both, liquid and ice crystal properties were derived? Please add this shortly to clarify to the reader.**

*AR: Absolutely. This has been specified line 289: "For g ranging between 0.8 and 0.83, both liquid and ice properties are derived".*

**RC: L253: I don't think, that Table 3 is needed, when you properly describe the matrix in the text here. Except of the range g=0.8-0.83 everything was given in the text.**

*AR: this was a request from the editor.*

**RC: L254: I would suggest to place this essential sentence a bit earlier. Maybe at line 248.**

*AR: This has been done. (Lines 262-264)*

**RC: L265-267: You don't have to repeat this. Remove sentence.**

*AR: Done*

**RC: L276: "droplet number size distribution" is the correct name of the quantity.**

*AR: Fixed.*

**RC: L297: Here only particle larger 100 _m are considered. But all other plots and analyzed parameters include particles less than 100 _m, e.g., number size distributions and D$_e$_.**
**Be consistent or give a reasons why not to be consistent.**

*AR: If there is a shattering effect, the number concentration is the parameter the most impacted. Additionally, all the ice particles are included when assess the other microphysical parameter in order to allow accurate comparison with previous work of* **(McFarquhar et al., 2011;  Jackson et al., 2012…)**

**RC: L305: Introduce acronyms ISDAC and MPACE.**

*AR: Done*

**RC: L305: A more detailed discussion with this available data would be helpful.**

*AR: The following text has been added at lines 397-407:*

*"These studies were based on 53 cloud profiles during the M-PACE campaign (**McFarquhar et al., 2011**) and 41 cloud profiles during the ISDAC campaign (**Jackson et al., 2012**). The ice crystal properties of single layer MPCs observed over the Beaufort Sea region did not show any significant vertical variability.*

*Typical IWC and particle concentration (for crystals with size larger than 125µm) values lied between 0.006 and 0.025 g.m-3 and between 1.6 L-1 and 5.6 L-1 for the M-PACE situations. These values are similar to those of the COLD and WARM_NO cases of the present study. Averaged values of IWC and particle concentration during ISDAC are in the range of the WARM_SO situations of the present work with values around 0.02 g.m-3 and 0.27 L-1 respectively for the ISDAC situations. The average ice crystal size observed during M-PACE is around 50 µm which is smaller than the typical size found in our study. It could be explained by less efficient WBF and riming processes and smaller droplet number also observed during M-PACE."*

**RC: L319: In this analysis the data of the four campaigns are merged. Is there evidence for differences in the particle shapes considering the strong temperature difference during ASTAR 2007?**

*AR: As shown in Figure R1 below, differences may be seen according to the temperature. The warm regimes (right panel, WARM_NO and WARM_CO have been merged) show the presence of some large droplets, which are not present in the cold regime (left panel).*

*Moreover, the plates/dendrites particles are clearly present below -10 °C both at the warm and cold regimes, and around -4°C in the warm regimes. Furthermore, this feature is consistent with the snow crystal morphology diagram (**Libbrecht, 2005; Nakaya, 1954**).*

*This has been mentioned in the discussion (section 3.3, lines 489-493) as follow:*

*"Moreover, the habit classification as a function of the temperature shows differences between the COLD regime and the WARM regimes (not shown here). This concern in particular the presence of some large droplets in the WARM regimes which are not present in the COLD regime, and the presence of plate and stellar particles below -10°C or around -4°C, which is consistent with the classical ice crystal morphology diagram ((**Libbrecht, 2005; Nakaya, 1954**)."*

[Figure]

**Figure R1: Particle shape distribution (in mass) according to the temperature for the COLD regime (left) and for the WARM_NO and WARM_SO regimes (right)**

**RC: L337: add "at low altitudes" for "...MPC in the Svalbard region at low latitudes."**
*AR: Done*

**RC: L345: This section analyses phase function and g and introduced these quantities. However, g values were already extensively used in the sections before to discriminate the cloud particle phase. The presentation of g and discussion has to be given first. By presenting g and phase function at this position of the manuscript, the analysis does not give any further new results and insight. The general location of ice and liquid particles was already discussed before. I therefore, recommend to shift this discussion somewhere earlier, before the first g values are used.**
*AR: This section has been moved and included in section 2.3 where the method for the phase discrimination is described.*

**RC: L374: I would also consider the sampling method to cause a mixing or averaging at cloud top. Assuming a cloud with variable cloud top altitude (even if only a couple of meters) and the aircraft ascending/descending through cloud top with a certain horizontal speed. If the ascending/descending rate is to low, then different parts of the horizontal inhomogeneous cloud top are averaged. This automatically will lead to lower LWC when averaging cloudy and cloud free patches.**
*AR: We agree with the reviewer's comment. The following sentence has been added to the manuscript (lines 454 to 456):* "Additionally, the data collected in this part of the cloud may also lead to a slight underestimation of the LWC since a mixing of cloudy and cloud free patches could be averaged together given the sampling resolution (i.e. 100 m)."

**RC: L376: Typo: "an" ice crystal growth...**
*AR: Fixed*

**RC: L382: What about vertical wind speed? Was this measured and can this be included in the analysis. Especially for such microphysical processes, the updraft velocity is important (Korolev, A. and P.R. Field, 2008: The Effect of Dynamics on Mixed-Phase Clouds: Theoretical Considerations. J. Atmos. Sci., 65, 66–86, doi: 10.1175/2007JAS2355.1.)**
*AR: We agree that the vertical velocity is a parameter of primary importance to the study of microphysical processes, but it has not been measured for these campaigns. This sentence has been added lines 464-465:* "Measurements of vertical wind speed (which are not available for these campaigns) would be helpful to confirm this hypothesis."

**RC: L383: LWC: Shortly explain how this was calculated. On what measured data the calculation is based?**
*AR: The theoretical adiabatic LWC is calculated assuming a non-entraining parcel of moist air rising and becoming saturated and is calculated from the measured values of p and T from cloud base to cloud top during the ascents and descents of the aircraft. The following equations are used:*

$$LWC_{adiab} = \rho_{air}\, q_{liq}(p,T)$$

*where $q_{liq}(p,T)$ is the density ratio of liquid to air (liquid mixing ratio) at pressure p and temperature T.*
*It is determined for each level z in the cloud according to:*

$$q_{liq}(z) = q^{sat}_{vap}(p_{base}, T_{base}) - q^{sat}_{vap}(p(z),T(z)) \text{ with :}$$

$p_{base}, T_{base}$ : pressure and temperature at cloud base
$p(z),T(z)$ : pressure and temperature at level z in the cloud
$q^{sat}_{vap}$ : saturation vapor mixing ratio.

*The following text has been added lines 466-468:*
*"Theoretical adiabatic LWC has also been determined assuming a non-entraining parcel of moist air rising and reaching saturation. It is calculated from the pressure and temperature measurements from cloud base to cloud top"*

**RC: L392: In what there is coherence with previous work? Such unspecific statements occur more often throughout the manuscript. It is always mentioned, that there is agreement, but it is not discussed in detail, what i particular agrees. Please look though the manuscript and be more specific.**
*AR: All along the discussion, specific statements have been added concerning the consistence of our results with the previous studies, taking into account the classification according to the meteorological conditions.*

**RC: L466: This is a repetition. It was just written before. Remove the whole introduction of 4. From a modeling perspective also other parameter are needed to be correlated with the cloud properties. But these are not adressed here because no measurements are available. So I would not start with this motivation...**
*AR: The needs for the modeling have been now described in the introduction of the revised manuscript.*

**RC: L484: How does this compare to standard models of the relationships between optical thickness and LWP. Actually these standard model equations include the particle size, which was not done here.**
*AR: The particle size is implicitly present in the IWC-Extinction relationships as it is linked to the ratio of IWC (or LWC) on extinction coefficient. The models commonly used IWP (or LWP) – optical thickness relationships, which is a similar way, but by integrating IWC and extinction over the altitude.*

**RC: L489: The upper range is missing for the temperature range of this study.**
*AR: this has been fixed.*

**RC: L492: "very few previous studies": There are plenty of studies analyzing the properties in Arctic MPC although many use ground based remote sensing (Shupe, De Boer, etc). Some of them are also given in the test below.**
*AR: We meant that very few studies exist concerning the determination of these properties from in situ measurements and in this region. This has been specified in the text line 589: "...from in situ measurements in this region of the Arctic."*

**RC: L503: What is the hypothesis in linking cloud top temperature and IWP, LWP?**
*AR: The cloud top temperature is the coldest temperature and a controlling parameter for ice initiation. Moreover, it allows to be consistent and useful with passive measurements which measured a cloud top temperature*

**RC: L516-525: This paragraph is rather part for the motivation. Here in the main text results should be presented and discussed. Add this discussion (state of the art microphysical**

**processes...) to the introduction.**
*AR: Done*

**RC: L529: Give the uncertainties for the fit parameter. There are statistical means to calculate these. Only if the uncertainty of the fit parameter covers the parametrization by Meyers et al. (1992) you can judge if both data sets agree with each other.**
*AR: The mean absolute error (MAE) has been calculated and displayed on the figure in order to estimate the uncertainties of the parameterization. The results show that the* **Meyers et al. (1992)** *parameterization is within the range of the uncertainties of our fitting.*
*The parameterization of* **Cooper (1986)** *and* **Young et al. (2017)** *have been added to complete the discussion.*
*All the discussion has been updated in lines 621 to 649.*

**RC: L531: Remove the last sentence. This is more useful for the conclusions and outlook.**
AR: Done

**RC: L537: rewrite: "...liquid fraction of individual measurements can be...".**
**The liquid fraction of an entire MPC would be LWP/TWP. Differentiate both parameters of liquid fraction for indivudual measurements and for the entire cloud. Spend symbols for both and introduce carefully.**
*AR: The entire liquid water fraction (LWP/TWP) is now noted $LWF_{total}$.*

**RC: L570: That first has to be shown! Especially with the good agrement with available parametrization, what is in particular the benefit of the new data set?**
*AR: The similarities with the previous parameterizations is shown on the figures by plotting the past relationships along with those from the present work. The benefit of the new data set is that it concerns Arctic MPC over the Atlantic side, compared to the previous works which deals with MPC over the Western arctic regions.*

**RC: L571-575: This fits better into the motivation. Move to introduction.**
*AR: Done*

**RC: L760: Conclusions: Of course it is a question of style. But if you have such long conclusions I see no benefit in putting these into a numeration. It could also be presented by regular paragraphs.**
*AR: We decided to keep this style.*

**RC: L762-765: The first two sentences are not a conclusion. Try to shorten and focus your conclusions.**
*AR: These two sentences have been removed.*

**RC: L792: This first has to be shown. You only presented the parameterizations but did not apply to remote sensing nor modelling.**
*AR: Applying these parameterizations to remote sensing nor modelling is beyond the scope of this paper, but is a future step to achieve. The following sentence has been added as a perspective (lines 742-743):*
*"A next step to the present work will be to apply the proposed parameterizations to remote sensing algorithms and modelling to investigate their relevance."*

**RC: L802: "Globally"? You just looked at 4 flights.**
*AR: This conclusion has been removed since it refers to the satellite validation section.*

**RC: L1030-1033: Reference is duplicated.**

*AR: It has been fixed.*

**RC: Figure 1: It would help if you indicate where the single profiles have been collected. The total flight paths are more or less irrelevant as they do not show where the profiles are measured.**

*AR: Figure 1 has been modified according to the reviewer's comment.*

**RC: Figure 3 and 4: Figures and labels are way to small! Even zooming in to 400%, I can not read the label. Spend more space for these figures. Maybe a 2x2 arrangement would help.**

*AR: Done*

**RC: Figure 5: The liquid droplets shown here are measurements from the FSSP? Or do you found liquid droplets larger than 100 _m with the CPI?**

*AR: some isolated droplets larger than 100µm can be found, but this large proportion is due to bad statistics in the 0 to -3°C temperature range (see answer to the following comment). Confusion with rimed particles may also occur, in particular due to the almost round shape of some rimed particles which may be confounded with drops by the classification algorithm.*

**RC: Figure 5: Between 0 and -3_C almost only liquid droplets are found. Is this liquid also the liquid droplets at cloud top in the upper panels? May be only measured above the inversions? Or is this smaller fraction of liquid and cloud top compared to high temperatures only due to bad statistics?**

*AR: This is due to bad statistics in the bottom of the WARM profiles. This temperature domain has been removed on the figure for clarity.*

**RC: Figure 6: This plot shows an average over all campaigns. Are there differences between the campaigns? I would assume similar differences compared to the LWC and IWC profiles.**

AR: Yes, difference are highlighted according to the different regimes, as shown in the Figure R2 below. However, this section has been moved in the section 2.4 concerning the experiment and the data processing in order to describe the discrimination of ice and liquid phases from the PN measurements. So we decided to keep this plot and the average over all the campaigns in this section.

[Figure]

**Figure R2: Scattering phase function according to the normalized altitude levels for the COLD, WARM_NO and WARM_SO regimes in the left, middle and right panels respectively.**

**RC: Figure 9: I'm not sure, if the log-scales somehow automatically gives a "good" fit. In linear space larger differences might be visible.**

*AR: The figure has been modified and is now in linear scale. Effectively, differences are more visible. The mean absolute errors (MAE) have been calculated to estimate the uncertainty of the parameterization. Values of MAE are displayed in the figure along with the fitting equations.*

**RC: Figure 9: Typo: Write "LWC" for the lower equation, liquid droplets.**
*AR: Done*

**RC: Figure 9: x-title: write "extinction coefficient"**
*AR: Done.*

**RC: Figure 10: In Fig. 8 cloud top temperature was given on the x-axis, here on y-axis. Make this consistent.**
*AR: Done.*

**RC: Figure 11: Panel a: Here the campaigns are not separately displayed. Why?**
*AR: Because the goal of using normalized altitudes is to merge the 4 data sets.*

**RC: Figure 11: Panel b and c: How there can be error bars beyond 100%?**
*AR: This is due to the method to calculate the uncertainties (for example, 25% of error for a ratio of 95% will give values larger than 100%). But the reviewer is right: values greater than 100% are not physically consistent. In the revised manuscript, we only consider maximum values up to 100%.*

**References**

Cooper, W. A.: Ice Initiation in Natural Clouds, Meteorol. Monogr., 21(43), 29–32, doi:10.1175/0065-9401-21.43.29, 1986.

Field, P. R., Hogan, R. J., Brown, P. R. A., Illingworth, A. J., Choularton, T. W., Kaye, P. H., Hirst, E. and Greenaway, R.: Simultaneous radar and aircraft observations of mixed-phase cloud at the 100 m scale, Q. J. R. Meteorol. Soc., 130(600), 1877–1904, doi:10.1256/qj.03.102, 2004.

Jackson, R. C., McFarquhar, G. M., Korolev, A. V., Earle, M. E., Liu, P. S. K., Lawson, R. P., Brooks, S., Wolde, M., Laskin, A. and Freer, M.: The dependence of ice microphysics on aerosol concentration in arctic mixed-phase stratus clouds during ISDAC and M-PACE, J. Geophys. Res., 117(D15207), doi:10.1029/2012JD017668, 2012.

Jourdan, O., Mioche, G., Garrett, T. J., Schwarzenböck, A., Vidot, J., Xie, Y., Shcherbakov, V., Yang, P. and Gayet, J.-F.: Coupling of the microphysical and optical properties of an Arctic nimbostratus cloud during the ASTAR 2004 experiment: Implications for light-scattering modeling, J. Geophys. Res., 115(D23206), doi:10.1029/2010JD014016, 2010.

Korolev, A. and Isaac, G. A.: Relative Humidity in Liquid, Mixed-Phase, and Ice Clouds, J. Atmospheric Sci., 63(11), 2865–2880, doi:10.1175/JAS3784.1, 2006.

Korolev, A. V., Isaac, G. A., Cober, S. G., Strapp, J. W. and Hallett, J.: Microphysical characterization of mixed-phase clouds, Q. J. R. Meteorol. Soc., 129(587), 39–65, doi:10.1256/qj.01.204, 2003.

Libbrecht, K. G.: The physics of snow crystals, Rep. Prog. Phys., 68(4), 855–895, doi:10.1088/0034-4885/68/4/R03, 2005.

McFarquhar, G. M., Zhang, G., Poellot, M. R., Kok, G. L., McCoy, R., Tooman, T., Fridlind, A. and Heymsfield, A. J.: Ice properties of single-layer stratocumulus during the Mixed-Phase Arctic Cloud Experiment: 1. Observations, J. Geophys. Res., 112(D24201), doi:10.1029/2007JD008633, 2007a.

McFarquhar, G. M., Zhang, G., Poellot, M. R., Kok, G. L., McCoy, R., Tooman, T., Fridlind, A. and Heymsfield, A. J.: Ice properties of single-layer stratocumulus during the Mixed-Phase Arctic Cloud Experiment: 1. Observations, J. Geophys. Res., 112(D24201), doi:10.1029/2007JD008633, 2007b.

McFarquhar, G. M., Ghan, S., Verlinde, J., Korolev, A., Strapp, J. W., Schmid, B., Tomlinson, J. M., Wolde, M., Brooks, S. D., Cziczo, D., Dubey, M. K., Fan, J., Flynn, C., Gultepe, I., Hubbe, J., Gilles, M. K., Laskin, A., Lawson, P., Leaitch, W. R., Liu, P., Liu, X., Lubin, D., Mazzoleni, C., Macdonald, A.-M., Moffet, R. C., Morrison, H., Ovchinnikov, M., Shupe, M. D., Turner, D. D., Xie, S., Zelenyuk, A., Bae, K., Freer, M. and Glen, A.: Indirect and Semi-direct Aerosol Campaign: The Impact of Arctic Aerosols on Clouds, Bull. Am. Meteorol. Soc., 92(2), 183–201, doi:10.1175/2010BAMS2935.1, 2011.

Meyers, M. P., DeMott, P. J. and Cotton, W. R.: New Primary Ice-Nucleation Parameterizations in an Explicit Cloud Model, J. Appl. Meteorol., 31(7), 708–721, doi:10.1175/1520-0450(1992)031<0708:NPINPI>2.0.CO;2, 1992.

Mioche, G., Jourdan, O., Ceccaldi, M. and Delanoë, J.: Variability of mixed-phase clouds in the Arctic with a focus on the Svalbard region: a study based on spaceborne active remote sensing, Atmospheric Chem. Phys., 15(5), 2445–2461, doi:10.5194/acp-15-2445-2015, 2015.

Nakaya, U.: Snow Crystals: Natural and Artificial, Cambridge Harvard University Press., 1954.

Rangno, A. L. and Hobbs, P. V.: Ice particles in stratiform clouds in the Arctic and possible mechanisms for the production of high ice concentrations, J. Geophys. Res., 106(D14), 15065, doi:10.1029/2000JD900286, 2001.

Young, G., Jones, H. M., Choularton, T. W., Crosier, J., Bower, K. N., Gallagher, M. W., Davies, R. S., Renfrew, I. A., Elvidge, A. D., Darbyshire, E., Marenco, F., Brown, P. R. A., Ricketts, H. M. A., Connolly, P. J., Lloyd, G., Williams, P. I., Allan, J. D., Taylor, J. W., Liu, D. and Flynn, M. J.: Observed microphysical changes in Arctic mixed-phase clouds when transitioning from sea ice to open ocean, Atmospheric Chem. Phys., 16(21), 13945–13967, doi:10.5194/acp-16-13945-2016, 2016.

Young, G., Connolly, P. J., Jones, H. M. and Choularton, T. W.: Microphysical sensitivity of coupled springtime Arctic stratocumulus to modelled primary ice over the ice pack, marginal ice, and ocean, Atmospheric Chem. Phys., 17(6), 4209–4227, doi:10.5194/acp-17-4209-2017, 2017.